# SₑRA: Self-Reviewing and Alignment of LLMs using Implicit Reward Margins

**Jongwoo Ko**[1*]   **Saket Dingliwal**[2]   **Bhavana Ganesh**[2]   **Sailik Sengupta**[3]
**Sravan Bodapati**[2]   **Aram Galstyan**[2]

[1]KAIST AI        [2]**a**‚mazon AGI        [3]**a**WS AI Labs

## Abstract

Direct alignment algorithms (DAAs), such as direct preference optimization (DPO), have become popular alternatives for Reinforcement Learning from Human Feedback (RLHF) due to their simplicity, efficiency, and stability. However, the preferences used in DAAs are usually collected before the alignment training begins and remain unchanged (off-policy). This design leads to two problems where the policy model (1) picks up on spurious correlations in the dataset (as opposed to learning the intended alignment expressed in the human preference labels), and (2) overfits to feedback on off-policy trajectories that have less likelihood of being generated by the updated policy model. To address these issues, we introduce Self-Reviewing and Alignment (**SeRA**), a cost-efficient and effective method that can be readily combined with existing DAAs. **SeRA** comprises of two components: (1) *sample selection* using implicit reward margins, which helps alleviate over-fitting to some undesired features, and (2) *preference bootstrapping* using implicit rewards to augment preference data with updated policy models in a cost-efficient manner. Extensive experimentation, including some on instruction-following tasks, demonstrate the effectiveness and generality of **SeRA** in training LLMs on offline preference datasets with DAAs.

## 1 Introduction

Large Language models (LLMs; Achiam et al. 2023; Team et al. 2023; Jiang et al. 2024) have shown mastery on a multitude of tasks in artificial intelligence (AI), ranging from creative writing (Wang et al., 2024b) to code generation (Li et al., 2023a), and mathematical reasoning (Ahn et al., 2024). With success, comes concerns related to their safety, reliability, and potential for misuse in sensitive domains like social manipulation, cyber-attacks, etc. To address some of these challenges, works have considered aligning LLMs to human values/preferences using approaches like Reinforcement Learning from Human Feedback (RLHF; Ouyang et al. 2022; Bai et al. 2022a).

Initial RLHF approaches, such as Proximal Policy Optimization (PPO), necessitated the need for two separate models— a policy model (the LLM model to be aligned) and a reward model (Schulman et al., 2017; Ouyang et al., 2022). Such approaches often result in challenges related to stability and scalability (Rafailov et al., 2023; Zhao et al., 2023; Llama Team, 2024). To mitigate these shortcomings, Direct Alignment Algorithms (DAAs), such as direct preference optimization (DPO; Rafailov et al. 2023), sequence likelihood calibration with human feedback (SLiC-HF; Zhao et al. 2023), and identity policy optimization (IPO; Azar et al. 2024), have emerged as popular alternatives. In DAA, we directly update the policy model/LLM using (some closed-from solution of) the pairwise preference data without the need for an explicit reward model, making the alignment process simpler, more efficient and stable compared to earlier methods (Rafailov et al., 2023).

However, DAAs leverage preference rating on off-policy trajectories collected prior to alignment tuning (at times, generated by a different LLM/policy-model (Zhao et al., 2023; Tunstall et al., 2023)), resulting in two challenges. *First*, the off-policy preference data, which still require labor-intensive annotations, often contain noisy features orthogonal to the true preference objective (e.g. longer responses are more preferred; Park et al. 2024). Thus, training on this data can teach the policy model to over-fit to such noise (Mitchell, 2023; Chowdhury et al., 2024), thereby learn spurious correlations (Park et al., 2024; Rafailov et al., 2023). While some works try to consider explicit

---

*Work done as a research intern at **a**‚mazon. Correspondence to jongwoo.ko@kaist.ac.kr

regularization to alleviate over-fitting to spurious features (Park et al., 2024), identifying all possible spurious features is often challenging and incomplete (Rafailov et al., 2024). *Second*, the preference feedback, being off-policy, cannot aid the policy model to obtain feedback on its own generations during alignment training. As the off-policy and the on-policy may not belong to the same distribution, this indirect feedback can inhibit improvement of the policy model (Tajwar et al., 2024). The latter shortcoming is partially addressed by PPO-like methods, where the stand-in reward model can rate on-policy generations (Guo et al., 2024).

To inherit the best of both worlds, several works have explored using RL from AI-Feedback (RLAIF; Guo et al. 2024; Rosset et al. 2024; Lee et al. 2023; Bai et al. 2022b). These approaches aim to mimic the human's preference rating behavior using high-quality LLMs (via API access). Unfortunately, this approximate preference-distillation-approaches are both inefficient and expensive. For example, Direct Nash optimization (DNO; Rosset et al. 2024) incurs a cost of $34,000 to obtain preference annotations using GPT-4 (Zheng et al., 2023). Thus, the motivations of bypassing training of a reward model lands up eventually incurring exorbitant costs.

**Contributions.** We propose to augment DAAs with a novel strategy, **S**elf-**R**eviewing and **A**lignment (`SeRA`) that uses Implicit Reward Margin (IRM; defined as Equation 5) for off-policy sample selection and iteratively bootstraps preference data for alignment. Specifically, we show that:

- **IRM-based Off-policy Sample Selection** helps mitigate over-optimization of policy models to spurious correlations (such as considering response length to gauge preference). Its efficacy and cost-effectiveness is consistent across various DAAs, datasets, problems, and model variants.
- **IRM-based Preference Data Bootstrapping** mitigates DAAs from continuously updating policy models with off-policy data and proposes a decoding and rejection sampling approach to extract informative policy pairs (based on IRM) for continual alignment training. This improves the efficacy and efficiency of DAAs without the need for expensive (& external) reward models.
- **Better Performance and Versatility:** We empirically showed that `SeRA` can be widely used across various DAAs (*e.g.,* DPO, IPO, SLiC-HF, SimPO) and on various LLMs (*e.g.*, TinyLlama-1.1B, Pythia-2.8B, Mistral-7B) consistently outperforming SoTA baselines (Kim et al., 2024a; Pattnaik et al., 2024). Finally, we conduct an exploratory analysis and ablations to better understand `SeRA`.

## 2 BACKGROUND

In this section, we provide a brief overview of related works (§2.1) and discuss some preliminaries (§2.2) necessary to understand our contribution. A more comprehensive discussion of related works is available in Appendix A.

### 2.1 RELATED WORK

Actor-critic RLHF frameworks (Christiano et al., 2017; Stiennon et al., 2020; Bai et al., 2022a; Ouyang et al., 2022) seeks to align language models to human preferences, but is often unstable during training and memory-intensive (requiring the policy model and reward model to be on device simultaneously). To mitigate this, several algorithms, such as direct preference optimization (DPO; Rafailov et al. 2023) and sequence likelihood calibration (SLiC-HF; Zhao et al. 2023), learn the contrastive preference in the offline setting using a closed-form loss function without the need for an critic/reward model. Azar et al. (2024) argued that without regularization, a policy can easily overfit to deterministic preferences and introduced identity preference optimization (IPO) to directly optimize offline preference probabilities with regularization. Moreover, the development of offline preference datasets like UltraFeedback (Cui et al., 2023) and OpenOrca (Lian et al., 2023) increases the practical accessibility of aligning LLMs.

Despite their effectiveness, offline approaches can suffer from distribution mismatch between the initial data and updated policy model (Guo et al., 2024; Ko et al., 2024). To address this, Guo et al. (2024) proposed online AI-feedback (OAIF) by using another LLM to annotate which of two online-sampled outputs from the current policy is preferred. Rosset et al. (2024) suggested direct Nash optimization (DNO), which is trained on iteratively updated preference pairs from the recent policy model based on its theoretical soundness. While effective, these methods incur high expenses for human labeling or proprietary LLM API calls, making them difficult to implement practically.

Another cost-efficient alternative is using policy models as preference labelers or leveraging implicit preference superiority with high-quality human responses. Self-rewarding LMs (Yuan et al., 2024)

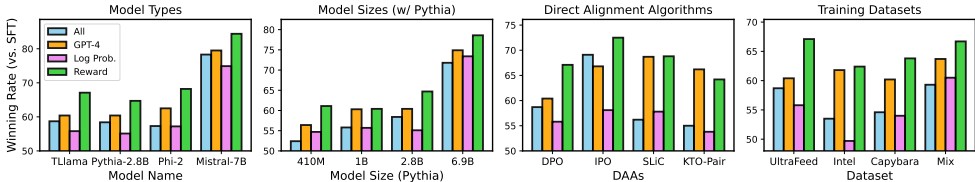

Figure 1: Motivation of **SeRA** : IRM-based selection showed higher win-rates (measured by Claude 3-as-a-Judge (Zheng et al., 2023; Anthropic, 2024)) against the base SFT model across a variety of settings– (a) Model types: TinyLlama (Zhang et al., 2024), Pythia-2.8B (Biderman et al., 2023), Phi-2 (Li et al., 2023c), and Mistral (Jiang et al., 2023). (b) Model size: various model sizes of Pythia (410M, 1.0B, 2.8B, 6.9B). (c) DAAs: DPO (Rafailov et al., 2023), IPO (Azar et al., 2024), SLiC (Zhao et al., 2023), and KTO-Pair (Ethayarajh et al., 2024) and (d) training datasets.

studied the benefits of iteratively training on preferences derived from recent policy's sampled outputs. However, in their work, they used the LLM itself as the annotator based on prompting (Zheng et al., 2023), which is only valid for LLMs capable of following the prompt properly. Self-play fine-tuning (SPIN; Chen et al. 2024b) and adversarial preference optimization (APO; Cheng et al. 2023) are both iterative LLM training techniques that are compatible with contrastive losses. However, they make the very limiting assumption that the high-quality predefined response is better than the policy generation, without considering any annotator feedback (Rosset et al., 2024).

## 2.2 PRELIMINARIES

**Reinforcement Learning From Human Feedback.** The goal of RLHF is to optimize the policy LLM $\pi_\theta$ such that it maximizes the expected value of the reward function. A common approach to modeling the reward function is using Bradley-Terry (BT; Bradley & Terry 1952) model:

$$p(\boldsymbol{y}_w \succ \boldsymbol{y}_l | \boldsymbol{x}) = \frac{\exp\left(r(\boldsymbol{x}, \boldsymbol{y}_w)\right)}{\exp\left(r(\boldsymbol{x}, \boldsymbol{y}_w)\right) + \exp\left(r(\boldsymbol{x}, \boldsymbol{y}_l)\right)} = \sigma(r(\boldsymbol{x}, \boldsymbol{y}_w) - r(\boldsymbol{x}, \boldsymbol{y}_l)), \quad (1)$$

where $\boldsymbol{y}_w$ (and $\boldsymbol{y}_l$) denotes the preferred/winning (and losing) policy, $p$ denotes the preference distribution that approximates an unobserved latent reward $r(\boldsymbol{x}, \boldsymbol{y})$, and $\sigma$ is the logistic function. To achieve this, RLHF first trains a reward model $r_\phi(\boldsymbol{x}, \boldsymbol{y})$. Then, RLHF updates $\pi_\theta$ with an on-policy RL algorithm like PPO (Schulman et al., 2017), iteratively optimizing the model to provide responses more preferred by human. The most common objective is

$$\mathcal{L}_{\text{RLHF}} = \mathbb{E}_{\boldsymbol{x} \sim \mathcal{D}, \boldsymbol{y} \sim \pi_\theta(\cdot | \boldsymbol{x})} \left[r_\phi(\boldsymbol{x}, \boldsymbol{y})\right] - \beta \mathbb{D}_{\text{KL}} \left[\pi_\theta(\boldsymbol{y}|\boldsymbol{x}) \| \pi_{\text{ref}}(\boldsymbol{y}|\boldsymbol{x})\right], \quad (2)$$

which enforces a KL divergence (Kullback & Leibler, 1951) penalty with a reference distribution $\pi_{\text{ref}}(\boldsymbol{y}|\boldsymbol{x})$, to prevent the LLM $\pi_\theta$ from straying too far from its initialization. The hyper-parameter $\beta$ balances between exploiting the reward function and deviating from $\pi_{\text{ref}}(\boldsymbol{y}|\boldsymbol{x})$.

**Direct Alignment Algorithms.** RLHF is computationally expensive and unstable (Rafailov et al., 2023; Llama Team, 2024). Thus, many algorithms (Rafailov et al., 2023; Zhao et al., 2023; Azar et al., 2024) have been proposed to overcome these challenges. A common idea is to analyti-cally derive the optimal policy and parameterize it using the reward function from Equation 2. In DPO (Rafailov et al., 2023), the optimal policy $\pi^*$ under the BT model satisfies:

$$p^*(\boldsymbol{y}_w \succ \boldsymbol{y}_l | \boldsymbol{x}) = \left(1 + \exp\left(\beta \log \frac{\pi^*(\boldsymbol{y}_l | \boldsymbol{x})}{\pi_{\text{ref}}(\boldsymbol{y}_l | \boldsymbol{x})} - \beta \log \frac{\pi^*(\boldsymbol{y}_w | \boldsymbol{x})}{\pi_{\text{ref}}(\boldsymbol{y}_w | \boldsymbol{x})}\right)\right)^{-1}, \quad (3)$$

where $p^*(\boldsymbol{y}_w \succ \boldsymbol{y}_l | \boldsymbol{x})$ is underlying true preference, the probability that $\boldsymbol{y}_w$ is more preferred than $\boldsymbol{y}_l$. With human preference data expressed in terms of the optimal policy rather than the reward model, we can formulate a maximum likelihood objective for a parameterized policy $\pi_\theta$.

$$\mathcal{L}_{\text{DPO}}(\boldsymbol{x}, \boldsymbol{y}_w, \boldsymbol{y}_l) = -\log \sigma \left(\beta \log \frac{\pi_\theta(\boldsymbol{y}_w | \boldsymbol{x})}{\pi_{\text{ref}}(\boldsymbol{y}_w | \boldsymbol{x})} - \beta \log \frac{\pi_\theta(\boldsymbol{y}_l | \boldsymbol{x})}{\pi_{\text{ref}}(\boldsymbol{y}_l | \boldsymbol{x})}\right) \quad (4)$$

Current methods for LLM alignment first collect a dataset of pairwise preferences by obtaining two responses to an input prompt $\boldsymbol{x}$ generated using an LLM. Then, human or AI annotators rank these responses, yielding a preferred response $\boldsymbol{y}_w$ and a less preferred one $\boldsymbol{y}_l$.

## 3 OUR METHOD: SELF-REVIEWING AND ALIGNMENT

In contrast to recent works on DAAs (Rosset et al., 2024; Guo et al., 2024), we propose **SeRA** to improve alignment with DAA without the need for any external supervision. While self-verification highlights that a policy model can dual up as a reward model (Weng et al., 2022), we show that this self-supervision can be done implicitly for alignment using the notion of Implicit Reward Margin (IRM). After defining IRM in this section, we describe (1) an IRM-based off-policy sample selection, and (2) an IRM-based iterative preference data bootstrapping. Finally, we describe our approach **SeRA** and how it can be seamlessly incorporated into DAAs.

### 3.1 IMPLICIT REWARD MARGIN (IRM)

While works have shown that alignment datasets may have ambiguous preferences (Yang et al., 2023; Chowdhury et al., 2024), even the use of unambiguous preference data can result in policy models unintentionally learning spurious correlations (e.g. on response length) (Park et al., 2024; Rafailov et al., 2024). Recent works consider selecting preference samples $(\boldsymbol{x}, \boldsymbol{y}_w, \boldsymbol{y}_l)$ where there exists a large difference in preferences (i.e. $\boldsymbol{y}_w \succ\succ \boldsymbol{y}_l$) to help mitigate the noise in the learning (Yang et al., 2023; Rosset et al., 2024). We leverage this observation and introduce a Implicit Reward Margin (IRM) to quantify the difference between the preference over two responses using the policy model $\pi_\theta$ itself. Formally, we define IRM as follows where $\pi_{\mathrm{ref}}$ is the reference model:

$$m(\boldsymbol{x}, \boldsymbol{y}_w, \boldsymbol{y}_l) \coloneqq \frac{1}{\beta}\left(r(\boldsymbol{x}, \boldsymbol{y}_w) - r(\boldsymbol{x}, \boldsymbol{y}_l)\right) = \log\frac{\pi_\theta(\boldsymbol{y}_w|\boldsymbol{x})}{\pi_{\mathrm{ref}}(\boldsymbol{y}_w|\boldsymbol{x})} - \log\frac{\pi_\theta(\boldsymbol{y}_l|\boldsymbol{x})}{\pi_{\mathrm{ref}}(\boldsymbol{y}_l|\boldsymbol{x})}, \quad (5)$$

which follows from Equation 3 and Equation 4. Unlike Rafailov et al. (2023), we omit the normalization term for the reward $r(\boldsymbol{x}, \boldsymbol{y}) \coloneqq \log\left(\frac{\pi_\theta(\boldsymbol{y}|\boldsymbol{x})}{\pi_{\mathrm{ref}}(\boldsymbol{y}|\boldsymbol{x})}\right)$ for tractability.

### 3.2 IRM-BASED OFF-POLICY SAMPLE SELECTION

We note that a policy model $\pi_\theta$ can generate mis-calibrated rewards for individual trajectories (Panickssery et al., 2024), but we empirically study if the reward margin between preferred $(\boldsymbol{y}_w)$ *vs.* less-preferred trajectories $(\boldsymbol{y}_l)$ can provide relevant signals to choose samples for improving alignment. Finally, we propose to filter out training samples with *lower* IRMs. To motivate our choice, we compare against other sample selection methods in the Figure 1. IRM-based sample selection (in green) consistently leads to higher win rates against the SFT model, when compared to alignment on the entire preference samples (in sky-blue), or strong baseliense that using RLAIF with GPT-4 score (Zheng et al. (2023); in orange) or the log probability of $\pi_{\mathrm{ref}}$ (Pattnaik et al. (2024); in pink) for sample selection. These gains hold true across model types, model size, DAAs, and datasets. Moreover, methods that use GPT-4 to obtain reward signals for sampling (Rosset et al., 2024; Zheng et al., 2023) incur high-cost; in comparison, our sample selection strategy that leverages $\pi_\theta$ to compute IRM is inexpensive and more effective.

**Mitigating over-optimization to spurious correlation.** To understand why IRM selection benefits alignment, we first highlight that recent works have shown that DAAs prioritize features of the data based on their complexity and prevalence, often resulting in learning preference signals based on dimensions orthogonal to the alignment objective, such as the length of responses (Park et al., 2024; Rafailov et al., 2024). Thus, we empirically analyze if IRM's sample selection strategy can weed out such noisy examples, thereby preventing over-optimization. To this extent, we study various correlations on TinyLlama on the UltraFeedback dataset for the four aforementioned sample selection strategies.

In Figure 2, we can observe that the rewards of the model trained on IRM-based sample selection (fourth column) has the highest $R^2$ scores with GPT-4 score, while also having the lowest $R^2$ score with the response lengths. This indicates that the policy model is highly aligned with the preferences of GPT-4 but does not learn the spurious feature of response length. Even though the rewards of the model without any sample selection (first column) also have the highest correlation with GPT-4 score, their correlation with response length is also high, a result consistent with previous works (Park et al., 2024; Rafailov et al., 2024) highlighting that DAAs indeed risks over-optimization where the model can learn spurious correlations. These results showcase that the proposed IRM selection strategy can mitigate over-optimization problems to response length, while other selection

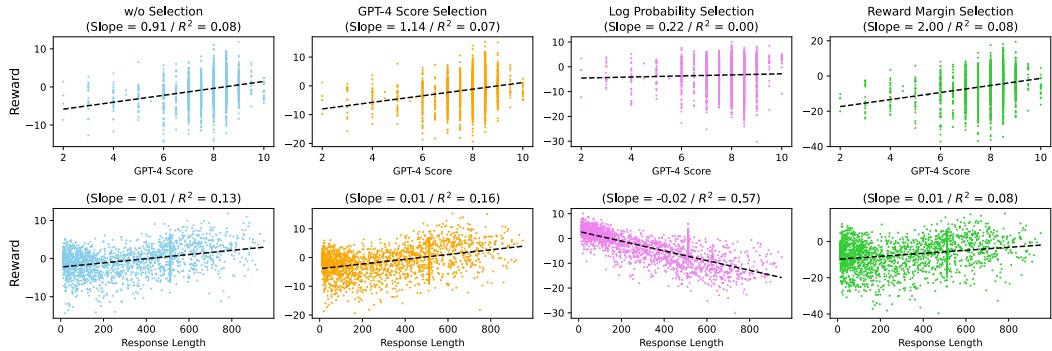

Figure 2: Comparison of correlations between the rewards of the trained policy model and features (*e.g.,* GPT-4 score and response length) across different selection methods (*i.e.,* no selection, GPT-4 score-based, log probability of the reference model, and IRM-based selection). **[Row 1]** Correlation between GPT-4 Score & implicit reward (*i.e,* $r(\mathbf{x}, \mathbf{y}_w)$) for $\mathbf{y}_w$. **[Row 2]** Correlation between response length (*i.e.* $|\mathbf{y}_w|$) and the implicit reward for chosen responses. The model with IRM selection (*i.e.* **[Column 4]**) shows the highest $R^2$ score for the first row, but the lowest $R^2$ score for the second row. This indicates that IRM-based selection strategy can effectively mitigate the over-optimization on response length (Park et al., 2024).

methods such as GPT-4 (second column; Rosset et al. 2024)[1] and log probability of the reference model (third column; Pattnaik et al. 2024) don't mitigate or, at times, exacerbate the undesired bias.

**Theoretical Intuition of Sample Selection.** Here, we additionally provide mathematical support for the empirical phenomena observed in Figure 2. Research has shown that DNNs first learn explicit features and then memorize minor ones (*e.g.,* spurious features, noisy labels) during the later training stage (Zhang et al., 2017; Liu et al., 2020; Nam et al., 2020; Liu et al., 2021). Inspired by this, we intend the policy LLM to first learn the human preference using confident preference pairs where true preferences $p^*(\mathbf{y}_w \succ \mathbf{y}_l | \mathbf{x}) \simeq 1$. As the estimated probability $\hat{p}_\theta(\mathbf{y}_w \succ \mathbf{y}_l | \mathbf{x}) = \sigma\left(\beta \log \frac{\pi_\theta(\mathbf{y}_w | \mathbf{x})}{\pi_{\text{ref}}(\mathbf{y}_w | \mathbf{x})} - \beta \log \frac{\pi_\theta(\mathbf{y}_l | \mathbf{x})}{\pi_{\text{ref}}(\mathbf{y}_l | \mathbf{x})}\right)$ is modeled by IRM in Eqn. (5) using the BT model, IRM selection removes samples with small margins that might lead to over-optimization on spurious features or noisy preferences.

**Remark 1** (**Informal Statement for Theorem 1**). *Under DPO training, with probability at least $1 - \delta$, the upper bound of the risk of $f(\mathbf{x}, \mathbf{y}_w, \mathbf{y}_l) := r(\mathbf{x}, \mathbf{y}_w) - r(\mathbf{x}, \mathbf{y}_l)$, denoted as $R(f)$, over the underlying distribution $\mathbb{P}$, where the sample set $S = (\mathbf{x}, \mathbf{y}_w, \mathbf{y}_l) \sim \mathbb{P}$, can be defined as:*

$$R(f) \leq \hat{R}(f; S) + \mathcal{O}\left(\mathbb{E}_{\mathbb{P}} \| \hat{p}_\theta(\cdot) - p^*(\cdot) \|_2\right) + \mathcal{O}\left(\sqrt{\tilde{\mathbb{V}}_{|S|}(f) \cdot \frac{\log(\mathcal{M}_{|S|}/\delta)}{|S|}} + \frac{\log(\mathcal{M}_{|S|}/\delta)}{|S|}\right),$$

*where $\mathcal{M}_{|S|}$ is uniform covering number. This reflects the risk of classifying preferable responses is upper bounded by the empirical risk over observed samples, the difference between estimated and true preferences. (See Appendix B for the proof.)*

Remark 1 indicates that training on clear preference samples reduces the upper bound of the risk compared to training on ambiguous samples. The inherent problem of DPO is that is treats all preference samples equally (*i.e.,* $p(\mathbf{y}_w \succ \mathbf{y}_l | \mathbf{x}) = 1, \forall (\mathbf{x}, \mathbf{y}_w, \mathbf{y}_l) \in S$) even when the true preferences are not, i.e. $p^*(\mathbf{y}_w \succ \mathbf{y}_l | \mathbf{x}) \neq p^*(\mathbf{y}'_w \succ \mathbf{y}'_l | \mathbf{x}')$. By minimizing the empirical risk on such datasets, the difference between $\hat{p}$ and $p^*$ increases, which can results in over-fitting to spurious features. However, by training LLMs on confident samples, we can achieve both lower empirical risk and less overfitting (*i.e.,* small differences between $\hat{p}_\theta$ and $p^*$). This theoretical interpretation supports the experimental observations shown in Figure 2. We further discuss each term in detail in Appendix B.

### 3.3  IRM-BASED PREFERENCE DATA BOOTSTRAPPING

As mentioned earlier, DAA's use of off-policy preference annotation suffers from distributional mismatch between the sequences observed during training (which are from different LLMs ahead of

---

[1]This result is consistent with recent works indicating that even GPT-4 evaluation has a length bias (Wu & Aji, 2023; Koo et al., 2023; Dubois et al., 2024a). This problem might also affect off-the-shelf reward models based on existing work on reward hacking of reward models (Gao et al., 2023), as shown in Table 3.

---

**Algorithm 1:** Self-Reviewing and Alignment

---

1: **Input**: Preference dataset $\mathcal{D}$, SFT model $\pi_{\theta_0}$, DAA loss function $\ell$
2: **Hyper-parameters**: # off-policy samples ($k$), # bootstrapped samples ($\tilde{k}$), Training epochs $T$
3: **Output**: Policy model $\pi_{\theta_T}$
4: **/\* Update policy model with offline dataset for $t = 1$ \*/**
5: Obtain $\pi_{\theta_1} \leftarrow \arg\max_\theta \mathbb{E}_{(\boldsymbol{x}, \boldsymbol{y}_w, \boldsymbol{y}_l) \in \mathcal{D}} \left[ \ell(\beta \log \frac{\pi_\theta(\boldsymbol{y}_w|\boldsymbol{x})}{\pi_{\theta_0}(\boldsymbol{y}_w|\boldsymbol{x})} - \beta \log \frac{\pi_\theta(\boldsymbol{y}_l|\boldsymbol{x})}{\pi_{\theta_0}(\boldsymbol{y}_l|\boldsymbol{x})}) \right]$
6: **for** $t \in \{2, \ldots, T\}$ **do**
7:     **Initialize** $\mathcal{M} \leftarrow \emptyset, \tilde{\mathcal{D}} \leftarrow \emptyset, \tilde{\mathcal{M}} \leftarrow \emptyset$
8:     **for** $(\boldsymbol{x}^{(i)}, \boldsymbol{y}_w^{(i)}, \boldsymbol{y}_l^{(i)}) \in \mathcal{D}$ **do**
9:         **/\* [§3.2] Compute IRM for Sample Selection \*/**
10:         $m^{(i)} := m_t(\boldsymbol{x}^{(i)}, \boldsymbol{y}_w^{(i)}, \boldsymbol{y}_l^{(i)})$             // Margin based on ensemble reward (Eq. 7)
11:         $\mathcal{M} \leftarrow \mathcal{M} \cup \{m^{(i)}\}$                 // Store IRM for each data-point
12:         **/\* [§3.3] IRM-based Preference Bootstrapping \*/**
13:         $\tilde{\boldsymbol{y}}_1, \ldots, \tilde{\boldsymbol{y}}_R \leftarrow \pi_{\theta_{t-1}}(\cdot | \boldsymbol{x}^{(i)})$          // Sample on-policy trajectories
14:         $\tilde{r}_{t,i,j} \leftarrow r_t(\boldsymbol{x}^{(i)}, \tilde{\boldsymbol{y}}_j) \; \forall j \in \{1, \ldots, R\}$      // Get implicit reward for trajectories (Eq 7)
15:         $\tilde{\mathcal{D}} \leftarrow \tilde{\mathcal{D}} \cup (\boldsymbol{x}^{(i)}, \tilde{\boldsymbol{y}}_{\arg\max_j \tilde{r}_{t,i,j}}, \tilde{\boldsymbol{y}}_{\arg\min_j \tilde{r}_{t,i,j}})$    // Select trajectories to maximize IRM
16:         $\tilde{\mathcal{M}} \leftarrow \tilde{\mathcal{M}} \cup \{\max_j \tilde{r}_{t,i,j} - \min_j \tilde{r}_{t,i,j}\}$          // Store maximum IRM
17:     **end for**
18:     **/\* Select samples with the highest IRM \*/**
19:     $\mathcal{M}, \tilde{\mathcal{M}} \leftarrow \text{sort}(\mathcal{M}, \text{revesed=True}), \text{sort}(\tilde{\mathcal{M}}, \text{revesed=True})$
20:     $\mathcal{D}_t \leftarrow \{(\boldsymbol{x}^{(i)}, \boldsymbol{y}_w^{(i)}, \boldsymbol{y}_l^{(i)}) \in \mathcal{D} \mid m^{(i)} \in \mathcal{M}[:k]\} \cup \{(\boldsymbol{x}^{(i)}, \tilde{\boldsymbol{y}}_w^{(i)}, \tilde{\boldsymbol{y}}_l^{(i)}) \in \tilde{\mathcal{D}} \mid \tilde{m}^{(i)} \in \tilde{\mathcal{M}}[:\tilde{k}]\}$
21:     **/\* Update policy model \*/**
22:     Obtain $\pi_{\theta_t} \leftarrow \arg\max_\theta \mathbb{E}_{(\boldsymbol{x}, \boldsymbol{y}_w, \boldsymbol{y}_l) \in \mathcal{D}_t} \left[ \ell(\beta \log \frac{\pi_\theta(\boldsymbol{y}_w|\boldsymbol{x})}{\pi_{\theta_{t-1}}(\boldsymbol{y}_w|\boldsymbol{x})} - \beta \log \frac{\pi_\theta(\boldsymbol{y}_l|\boldsymbol{x})}{\pi_{\theta_{t-1}}(\boldsymbol{y}_l|\boldsymbol{x})}) \right]$
23: **end for**

---

training) and those generated by the iteratively updated policy LLM (Arora et al., 2022; Ko et al., 2024), leading to reduced efficacy of DAAs (Guo et al., 2024). To address this, we present preference data bootstrapping that considers a decoding-time strategy to sample candidate pairs from the (updated) policy LLM, followed by a rejection sampling of pairs with low IRM.

To build a preference pair, we sample the $R \; (\geq 2)$ distinct candidate responses $\boldsymbol{y}_1^{(i)}, \ldots, \boldsymbol{y}_R^{(i)} \sim \pi_{\theta_{t-1}}(\cdot | \boldsymbol{x}^{(i)})$ from a query $\boldsymbol{x}^{(i)}$ by using decoding-time sampling. We then compute the implicit reward for each response $j (\in \{1, \ldots, R\})$ using the term $r(\boldsymbol{x}^{(i)}, \boldsymbol{y}_j^{(i)})$ in Equation 5 and select the pair that maximizes the IRM.

Our method bears similarity to self-rewarding LMs (SRLM; Yuan et al. 2024) that reward responses by using LLM-as-a-Judge (Zheng et al., 2023). Unfortunately, this approach works reliably only when one uses LLMs with sufficient instruction-following capability (see Table 5), which leads to significant costs and limited applicability for smaller LLMs. In contrast, our approach is versatile across LLMs with a wide range of capacities and boasts strong empirical efficacy (see Table 6).

### 3.4 **SeRA**: SELF-REVIEWING AND ALIGNMENT

We first describe how our two proposed components can be incorporated in an iterative manner with DPO and then extend it for other DAAs. Algorithm 1 provides an overview of **SeRA**. As our method leverages the policy model from previous iterations both for sample selection (§3.2) and preference data bootstrapping (§3.3), we consider an iterative approach to alignment training. Similar to recent works (Kim et al., 2024a; Pattnaik et al., 2024), we also update the reference model $\pi_{\text{ref}}$ to the latest updated policy model after every iteration. At iteration $t$, we obtain $\theta_t$ using the following update step (instead of Equation 4 used in DPO):

$$\theta_t \leftarrow \arg\max_\theta \mathbb{E}_{(\boldsymbol{x}, \boldsymbol{y}_w, \boldsymbol{y}_l) \sim \mathcal{D}_t} \left[ -\log \sigma \left( \beta \log \frac{\pi_\theta(\boldsymbol{y}_w|\boldsymbol{x})}{\pi_{\theta_{t-1}}(\boldsymbol{y}_w|\boldsymbol{x})} - \beta \log \frac{\pi_\theta(\boldsymbol{y}_l|\boldsymbol{x})}{\pi_{\theta_{t-1}}(\boldsymbol{y}_l|\boldsymbol{x})} \right) \right], \quad (6)$$

where $\mathcal{D}_t$ is a dataset obtained at the start of iteration $t$, by combining $k$ samples from off-policy dataset using §3.2 and $\tilde{k}$ bootstrapped samples using method §3.3 with the highest IRM. At $t = 0$,

we consider the reference model to be the SFT model (i.e. $\pi_{\theta_0} = \pi_{\text{SFT}}$) that we seek to align. Note that at $t = 1$, $\mathcal{D}_t$ consists only of the original offline samples, as no aligned model is available yet.

**Ensemble of Reward Margin across Different Iteration.** As `SeRA` depends on leveraging the policy model being trained instead of external reward model supervision, it risks introduction of undesired bias or reward hacking (Pan et al., 2024) that can manifest as part of the implicit rewards used to calculate IRM. To alleviate this, we apply $m_t(\boldsymbol{x}, \boldsymbol{y}_w, \boldsymbol{y}_l) \coloneqq (1/\beta) \cdot (r_t(\boldsymbol{x}, \boldsymbol{y}_w) - r_t(\boldsymbol{x}, \boldsymbol{y}_l))$ rather than using Eqn. (5), where $r_t$ ($t \geq 3$) is defined as follows:

$$r_t(\boldsymbol{x}, \boldsymbol{y}) = (1 - \gamma) \log \frac{\pi_{\theta_{t-1}}(\boldsymbol{y}|\boldsymbol{x})}{\pi_{\theta_{t-2}}(\boldsymbol{y}|\boldsymbol{x})} + \gamma \log \frac{\pi_{\theta_{t-2}}(\boldsymbol{y}|\boldsymbol{x})}{\pi_{\theta_{t-3}}(\boldsymbol{y}|\boldsymbol{x})} = \log \frac{\pi_{\theta_{t-1}}(\boldsymbol{y}|\boldsymbol{x})^{(1-\gamma)}\pi_{\theta_{t-2}}(\boldsymbol{y}|\boldsymbol{x})^{\gamma}}{\pi_{\theta_{t-2}}(\boldsymbol{y}|\boldsymbol{x})^{(1-\gamma)}\pi_{\theta_{t-3}}(\boldsymbol{y}|\boldsymbol{x})^{\gamma}}, \quad (7)$$

where $\gamma$ is the ensemble coefficient. This is conducted both for preference pairs in the original datasets (§3.2) and generated samples (§3.3) to build $\mathcal{D}_t$ for iteration $t$. For $t < 3$, we use $r_t(\boldsymbol{x}, \boldsymbol{y}) = \log \frac{\pi_{\theta_{t-1}}(\boldsymbol{y}|\boldsymbol{x})}{\pi_{\theta_{t-2}}(\boldsymbol{y}|\boldsymbol{x})}$. Empirically, this approach showed consistent effectiveness (§5; Figure 5a).

**Extension to Other DAAs.** Instead of BCE loss function of IRM (*i.e.,* $m(\boldsymbol{x}, \boldsymbol{y}_w, \boldsymbol{y}_l)$) in Eqn. (4), SLiC-HF (Zhao et al., 2023) minimizes a hinge loss function of IRM:

$$\mathcal{L}_{\text{SLiC-HF}}(\boldsymbol{x}, \boldsymbol{y}_w, \boldsymbol{y}_l) = \max\left(0, 1 - \beta\left(\log \frac{\pi_\theta(\boldsymbol{y}_w|\boldsymbol{x})}{\pi_{\text{ref}}(\boldsymbol{y}_w|\boldsymbol{x})} - \log \frac{\pi_\theta(\boldsymbol{y}_l|\boldsymbol{x})}{\pi_{\text{ref}}(\boldsymbol{y}_l|\boldsymbol{x})}\right)\right), \quad (8)$$

where $1/\beta$ acts as the margin of miss-classification (Guo et al., 2024). IPO (Azar et al., 2024) minimizes the square loss function of IRM:

$$\mathcal{L}_{\text{IPO}}(\boldsymbol{x}, \boldsymbol{y}_w, \boldsymbol{y}_l) = \left(\left(\log \frac{\pi_\theta(\boldsymbol{y}_w|\boldsymbol{x})}{\pi_{\text{ref}}(\boldsymbol{y}_w|\boldsymbol{x})} - \log \frac{\pi_\theta(\boldsymbol{y}_l|\boldsymbol{x})}{\pi_{\text{ref}}(\boldsymbol{y}_l|\boldsymbol{x})}\right) - \frac{1}{2\beta}\right)^2 \quad (9)$$

Note that IPO and SLiC-HF, in contrast to DPO, doesn't assume a preference model like BT. Despite of such independence, the objective functions still contain the IRM term enabling us to easily extend `SeRA` to these DAAs. Thus, we use `SeRA` with these DAAs for our upcoming experiments.

## 4 EXPERIMENTS

### 4.1 EXPERIMENTAL SETUP

**Setup.** Our setup resembles Tunstall et al. (2023) and Hong et al. (2024), where they consider three models– TinyLlama (Zhang et al., 2024), Pythia-2.8B (Biderman et al., 2023), and Mistral-7B (Jiang et al., 2023). Initially, these models are fine-tuned on UltraChat-200k (Tunstall et al., 2023) followed by an alignment with DAAs and preference pairs from UltraFeedback. We use the binary version of UltraFeedback (which contains two response pairs with corresponding ratings for a given input query) (Tunstall et al., 2023) for all DAAs except Curri-DPO (Pattnaik et al., 2024), where we use the original UltraFeedback (that contains four response pairs) (Cui et al., 2023). In addition, we will also showcase `SeRA`'s prowess on other popular alignment datasets like HH-RLHF (Bai et al., 2022a) and TL;DR (Stiennon et al., 2020). A detailed description of the datasets and the implementation can be found in Appendix C. For all experiments, we set $T = 3$, $\gamma = 0.3$. If $N$ represent the total number of preference pairs in the offline dataset, we choose our values of $k, \tilde{k}$ s.t. $k + \tilde{k} = N$, allowing us a fair comparison against other baseline in terms of sample size. Based on a validation set, we set $k = 0.7N$, $\tilde{k} = 0.3N$ (labelled as `SeRA` in our tables). However, to showcase the full potential of our proposed method, we relax the sum-constraint and also report numbers for **best**-`SeRA`, where $k = 0.7N$, $\tilde{k} = 2N$.

**Baselines.** We compare `SeRA` to several previous DAAs. First, we consider DPO (Rafailov et al., 2023) and its variants: Iterative DPO (Kim et al., 2024a; Rosset et al., 2024), Curri-DPO (Pattnaik et al., 2024), SPIN (Chen et al., 2024b), and ORPO (Hong et al., 2024). Then, we use `SeRA` with IPO (Azar et al., 2024) and SLiC-HF (Zhao et al., 2023) and compare against the original method and its iterative version. For experiments on synthetic noisy preference datasets, we also include cDPO (Mitchell, 2023) and rDPO (Chowdhury et al., 2024).

**Evaluation.** We consider four popular test benchmarks – Alpaca bench (Dubois et al., 2024b), Vicuna bench (Chiang et al., 2023), Evol Instruct (Xu et al., 2023) test set, and the UltraFeedback (Cui et al., 2023; Tunstall et al., 2023) test set. For metrics, we use Claude 3 (Anthropic, 2024) as a judge to evaluate the quality of the generated response and report win rate compared to another model.

Table 1: Comparison of performance where TinyLlama-1.1B, Pythia-2.8B, and Mistral-7B are fine-tuned on UltraChat-200k and preference pairs are sampled from UltraFeedback dataset with DPO (Rafailov et al., 2023) or its variants (Pattnaik et al., 2024; Chen et al., 2024b; Hong et al., 2024). Single represents average from single answer grading prompt while SFT, G3.5, G4 indicate the pairwise comparison (Zheng et al., 2023) between SFT model, `text-davinci-003`, and `gpt4_turbo`. The best and second best performances are highlighted **bold** and underline. ‡ indicate results from original models shared on HuggingFace.

| Model | Size | Technique | Alpaca Eval | | | | Vicuna Eval | | Evol-Instruct | | UltraFeed | |
|---|---|---|---|---|---|---|---|---|---|---|---|---|
| | | | Single | SFT | G3.5 | G4 | Single | SFT | Single | SFT | Single | SFT |
| TinyLlama | 1.1B | SFT | 5.41 | - | 28.9 | 1.7 | 6.05 | - | 5.48 | - | 4.98 | - |
| | | DPO | 5.82 | 55.1 | 33.5 | 2.9 | 6.31 | 57.5 | 5.50 | 56.0 | 5.34 | 59.8 |
| | | Iterative DPO | 6.08 | 64.5 | 36.1 | 3.2 | 7.09 | 77.5 | 6.03 | 62.4 | 5.49 | 64.4 |
| | | Curri-DPO | 6.16 | 62.3 | 40.7 | 2.8 | 6.85 | 77.5 | 6.00 | 61.5 | 5.63 | 66.8 |
| | | **SeRA**-DPO | **6.58** | **74.2** | 48.7 | 5.1 | **7.55** | **85.8** | **6.40** | **72.3** | 5.92 | 73.1 |
| | | **best-SeRA**-DPO | 6.47 | 70.0 | 49.1 | 6.1 | 7.31 | 82.5 | 6.18 | 70.0 | **6.06** | **79.5** |
| Pythia | 2.8B | SFT | 5.34 | - | 28.1 | 2.2 | 6.19 | - | 5.73 | - | 5.09 | - |
| | | DPO | 5.67 | 53.0 | 31.6 | 2.9 | 6.30 | 50.0 | 6.02 | 58.7 | 5.38 | 57.5 |
| | | Iterative DPO | 6.05 | 63.2 | 40.6 | 3.0 | 7.23 | 70.0 | 6.16 | 60.8 | 5.67 | 65.9 |
| | | Curri-DPO | 6.06 | 62.7 | 39.6 | 2.7 | 7.03 | 62.5 | 6.39 | 68.1 | 5.67 | 72.0 |
| | | **SeRA**-DPO | 6.40 | 70.7 | 50.2 | 6.0 | **7.59** | 76.8 | **6.68** | 74.2 | 5.93 | 71.7 |
| | | **best-SeRA**-DPO | **6.59** | **77.4** | **52.2** | **7.7** | 7.53 | **83.8** | 6.54 | **79.1** | **6.00** | **76.6** |
| Mistral | 7B | SFT‡ | 7.46 | - | 75.9 | 5.4 | 8.08 | - | 7.73 | - | 6.76 | - |
| | | DPO‡ | 7.37 | 71.4 | 74.9 | 17.5 | 8.39 | 83.8 | 7.15 | 61.7 | 6.42 | 65.6 |
| | | Iterative DPO | 8.35 | 89.0 | 93.2 | 19.7 | 8.74 | 88.2 | 8.49 | 87.8 | 7.56 | 81.8 |
| | | Curri-DPO | 8.28 | 87.6 | 93.1 | 13.8 | 8.75 | 92.5 | 8.45 | 85.8 | 7.52 | 80.8 |
| | | SPIN‡ | 7.64 | 65.1 | 77.1 | 8.4 | 8.40 | 70.0 | 7.66 | 61.7 | 6.63 | 61.0 |
| | | ORPO‡ | 8.29 | 84.1 | 91.6 | 14.5 | 8.82 | 83.8 | 8.61 | 86.0 | 7.58 | 78.8 |
| | | **SeRA**-DPO | 8.38 | 86.9 | 93.4 | 19.5 | 8.83 | 88.8 | 8.61 | 86.0 | 7.67 | 84.4 |
| | | **best-SeRA**-DPO | **8.56** | **92.7** | **94.8** | **27.3** | **8.94** | 91.3 | **8.68** | **90.1** | **7.90** | **90.6** |

Table 2: Comparison of performance where TinyLlama-1.1B and Pythia-2.8B are fine-tuned on UltraChat-200k and preference pairs are sampled from UltraFeedback dataset with IPO (Azar et al., 2024) and SLiC-HF (Zhao et al., 2023). The best performances are highlighted in **bold**.

| Method | Size | Technique | Alpaca Eval | | | | Vicuna Eval | | Evol-Instruct | | UltraFeed | |
|---|---|---|---|---|---|---|---|---|---|---|---|---|
| | | | Single | SFT | G3.5 | G4 | Single | SFT | Single | SFT | Single | SFT |
| TinyLlama | 1.1B | SFT | 5.41 | - | 28.9 | 1.7 | 6.05 | - | 5.48 | - | 4.98 | - |
| | | IPO | 5.94 | 54.1 | 29.3 | 2.7 | 6.84 | 63.8 | 5.91 | 50.7 | 5.55 | 64.7 |
| | | Iterative IPO | 5.98 | 56.5 | 30.1 | 2.2 | 6.94 | 63.8 | 6.00 | 59.2 | 5.65 | 64.7 |
| | | **SeRA**-IPO | **6.42** | **69.3** | **45.6** | **4.8** | **7.31** | **81.3** | **6.12** | **69.2** | **5.76** | **70.5** |
| | | SLiC | 6.19 | 65.9 | 40.1 | 3.5 | 7.27 | 71.3 | 6.05 | 65.1 | 5.57 | 66.8 |
| | | Iterative SLiC | 6.41 | 70.6 | **47.8** | 4.8 | 7.23 | **78.8** | 6.06 | 65.1 | 5.73 | 69.1 |
| | | **SeRA**-SLiC | **6.50** | **70.8** | 47.2 | **5.4** | **7.58** | 76.3 | **6.06** | **67.9** | **5.88** | **72.8** |
| Pythia | 2.8B | SFT | 5.34 | - | 28.1 | 2.2 | 6.19 | - | 5.73 | - | 5.09 | - |
| | | IPO | 6.15 | 67.1 | 41.8 | 4.5 | 7.32 | 80.0 | 6.27 | 66.5 | 5.65 | 75.3 |
| | | Iterative IPO | 6.55 | 75.2 | 50.2 | 5.0 | 7.61 | 80.0 | 6.68 | 71.3 | 6.02 | 75.5 |
| | | **SeRA**-IPO | **6.71** | **78.5** | **53.6** | **6.1** | **7.74** | **83.8** | **6.83** | **72.7** | **6.14** | **78.7** |
| | | SLiC | 6.07 | 65.2 | 41.7 | 2.8 | 7.06 | 66.3 | 6.24 | 60.8 | 5.66 | 69.9 |
| | | Iterative SLiC | 6.38 | 72.3 | 47.1 | 5.6 | 7.54 | 83.8 | **6.48** | **73.4** | 5.84 | 70.0 |
| | | **SeRA**-SLiC | **6.57** | **74.3** | **52.9** | **6.2** | **7.62** | **85.0** | 6.47 | 71.1 | **6.03** | **74.3** |

The win-rate adjudication assigns a weight of 1 to wins, 0.5 to ties, and 0 to a loss. We also use a single answer grading prompt to provide a 1-10 rating for the responses generated by the model. The evaluation prompts are taken from Zheng et al. (2023) and provided in Appendix C.3 for reference.

## 4.2 RESULTS

**Combining with DPO.** We assess the general instruction-following abilities of LLMs by comparing the DPO-based preference alignment algorithms. Tab. 1 depicts (1) the instruction-following performance of both variants of **SeRA**, showing their superiority over their baselines across diverse LLMs and evaluation metrics, and (2) that increased preference bootstrapping is more effective for stronger LLMs (*e.g.,* Mistral-7B), as seen in the larger performance gap between variants. These results highlight the versatility of the proposed method across different experimental setups.

**Extension to Various DAAs.** To show that **SeRA** can be improve various DAAs, we report instruction-following performance of **SeRA** integrating with IPO (Azar et al., 2024) and SLiC-HF (Zhao et al., 2023) in Table 2. Similar to Table 1, **SeRA** consistently improved the performance of both IPO and SLiC-HF. Compared to the DPO and SLiC-HF cases, the performance improvement on when **SeRA** is used with IPO is relatively smaller. We note that IPO has been shown to effectively alleviate over-optimization in comparison to other DAAs (Rafailov et al., 2024); hence the gains ob-

tained by **SeRA** indicate other potential upsides. Indirectly, these results support our argument that **SeRA** addresses the challenge of over-optimization. We additionally provided the effectiveness of **SeRA** on different types of implicit reward, SimPO (Meng et al., 2024), in Appendix D.3.

**Robustness to Noisy Preferences.** To illustrate the effectiveness of our proposed **SeRA** on noisy annotations, we experimented with synthetically noised preference datasets. Similar to prior works (Chowdhury et al., 2024; Wang et al., 2024a), we consider the standard noise model and randomly swap the chosen and rejected responses. Our experiments evaluate two setups– 20% and 40% noise in the Ultra-Feedback dataset. On TinyLlama trained on each noise split, we highlight the performance of various robust DAAs in Figure 3. In most of the evaluation benchmarks, **SeRA** showed su-

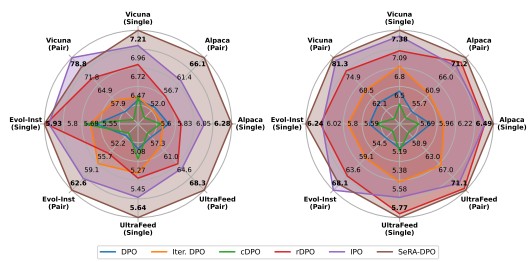

Figure 3: Comparison of performance on 20% **(left)** and 40% **(right)** of noisy preference

periority over other baselines. Moreover, the performance gap between DPO, iterative DPO, and **SeRA**-DPO is much larger than that on standard dataset. This indicate that our reward margin selection effectively filters-out noisy samples with swapped preferences.

**Efficacy on Different Preference Datasets.** To demonstrate the generality of **SeRA**, we also conducted experiments with TinyLlama on two other datasets: HH-RLHF (Ganguli et al., 2022) and TL;DR (Stiennon et al., 2020). Since these two datasets do not contain explicit GPT-4 scores for each response, we utilized a Mistral-7B-based reward model that achieves

Table 3: Results on HH-RLHF and TL;DR.

|  | DPO | IDPO | OAIF | **SeRA** $(\tilde{k}=0)$ | **SeRA** $(\tilde{k}=0.3N)$ |
|---|---|---|---|---|---|
| HH-RLHF | 56.7 | 63.0 | 64.6 | 65.4 | **66.1** |
| TL;DR | 59.7 | 59.6 | 59.5 | 61.0 | **62.7** |

high performance in RewardBench (Lambert et al., 2024) to implement online feedback from stronger LLMs (OAIF; Guo et al. 2024), which requires significantly more computational resources compared to ours. Further, we also showcase the importance of sample selection alone by benchmarking **SeRA** with $\tilde{k} = 0$. Table 3 shows that the effectiveness of our approach is not limited to any particular preference dataset. It overcomes problems like spurious correlations that practically exists in every preference dataset. Further, it can achieve this better performance at a lower cost.

## 5 ANALYSIS AND DISCUSSION

**Component Analysis.** In this section, we evaluate the efficacy of sample selection and preference bootstrapping in our proposed method. We plot the winning rate for different ratios of $k$ and $\tilde{k}$ in Figure 4. The solid lines represent results obtained by different iterations in **SeRA** with varying ratios of $k$ and $\tilde{k}$ maintaining $k + \tilde{k} = N$, while the dotted lines represent results obtained by using only IRM selection where $k = 0.7N, \tilde{k} = 0$. The results indicate that (1) IRM-based selection shows consistent improvement in comparison to using only offline samples (0% on x-axis), or only generated samples (100%), and (2) combining IRM-based sample selection and preference bootstrapping at moderate levels (*i.e.*, $30 - 70\%$) leads to consistent improvement, highlighting the importance of both components in **SeRA**. Without offline sam-

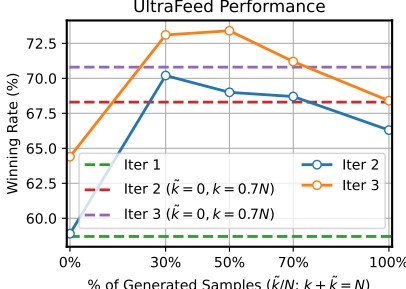

Figure 4: Pairwise comparison of models with varying percentage $k$ and $\tilde{k}$ while ensuring a fixed-sized training set (i.e. $\tilde{k}; k + \tilde{k} = N$).

ples, IRM and generated samples may lack valuable information for effective training, while relying solely on offline data can cause overfitting. Thus, the combination of selection and bootstrapping is crucial. Across iterations, this combination consistently outperforms solely using all offline samples, only generated samples, or IRM selection alone, confirming the importance and synergy of each component in **SeRA**.

**Analysis on Ensemble across Different Iterations.** We also investigate the efficacy of the ensemble across different iterations, reporting the performance and similarity of selected samples for

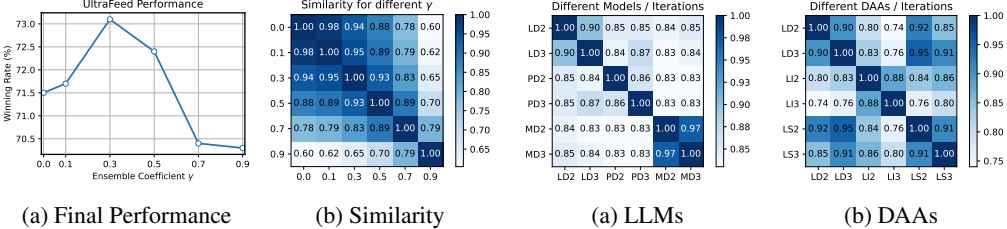

| (a) Final Performance | (b) Similarity | (a) LLMs | (b) DAAs |

Figure 5: The performance and similarity of se- Figure 6: The similarity of selected sample sets lected sample sets across different values of the between different LLMs or DAAs with training ensemble coefficient $\gamma$ in TinyLlama-1.1B. iterations.

the TinyLlama case, when combining IRM in the second and third iterations. Although we see high winning rate in Figure 5a, a moderate range of ensemble coefficients $\gamma$ (*i.e.*, 0.1–0.5) leads to greater performance improvements compared to higher coefficients (*i.e.*, 0.7–0.9). This suggests that the reward margin generated by the latter policy models play a more significant role. However, employing an ensemble of reward margins can effectively mitigate the deterioration caused by undesirable model bias. Additionally, Figure 5b examines the similarity of selected samples across different $\gamma$ values. The results show that mild levels of $\gamma$ only slightly alter the selected samples, with similarity scores ranging from 0.88 to 0.94. This slight modification is crucial as it ensures the retention of important samples while preventing undesired bias in the model.

**Selection Behavior.** We also checked the difference in the selected sample set with different training configurations such as DAAs, iterations, and LLMs. Figure 6a and Figure 6b depict the Jaccard similarity (Murphy, 1996) between two different selected sample sets from different LLMs and DAAs, respectively. All selected sets contain 70% of the samples from the original UltraFeedback dataset. For all axes, the first and second letters indicate the LLM (e.g., L, P, and M indicate TinyLlama, Pythia-2.8B, and Mistral-7B) and DAA (e.g., D, I, and S indicate DPO, IPO, and SLiC-HF), and the third number indicates the iteration count. In general, we observe high similarity in the selected samples across LLMs ($\geq 0.83$) and across DAAs ($\geq 0.74$). Across iterations, we also observe a slight variance in the samples selected (0.903 on avg.).

**More Preference Bootstrapping.** To uncover the full potential of **SeRA**, we train with more self-generated samples. We fix $k$ at $0.7N$. We compare the cases with $\tilde{k}$ ranging from 0 to $3N$. The results in Figure 7 illustrate that increasing the generated samples up to $2N$ leads to performance improvement, highlighting the efficacy of preference bootstrapping. However, we observe a minor drop at $3N$, possibly pointing to undesirable model bias or duplicate/less-diverse generated data. We hope to consider advanced decoding approaches Huang et al. (2024) to address the latter in the future.

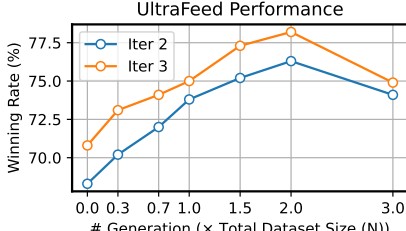

Figure 7: Pairwise comparison of models with fixed $k = 0.7N$ and varying $\tilde{k}$

## 6 CONCLUSION

In this work, we proposed **SeRA** that incorporates two components into Direct Alignment Algorithms (DAAs)– (1) an implicit reward margin-based sample selection of offline datasets mitigates over-optimization, supported by empirical evidence; and (2) a self-reviewed preference bootstrapping to enable direct feedback that considers the training-inference mismatch. In addition, we use a reward ensemble over policy models across iterations (that helps to develop a robust reward metric) to enable a high-quality dataset construction; the ensemble. Extensive experiments on diverse training configurations, across various LLMs, DAAs, and datasets, demonstrated the superior performance of **SeRA**. This highlights that even without additional supervision from humans or stronger LLMs, we can squeeze existing offline preference datasets further using policy model's metrics (e.g. reward margins) to improve the policy model. We believe that our work demonstrates the possibility of self-evolution of LLMs for achieving super-alignment (Burns et al., 2023) without supervision or with minimal supervision.

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

# Supplementary Material

## *SeRA*: *Self-Reviewing and Alignment of LLMs using Implicit Reward Margins*

## A    ADDITIONAL RELATED WORKS

Here, we further discuss additional related works regarding our proposed method, **SeRA**. Specifically, we focus on sample selection approaches and bootstrapping in the context of previous deep learning research, rather than preference optimization of LLMs.

**Over-optimization in LLM Alignment.**    Gao et al. (2023) defined over-optimization as the phenomenon where optimizing a model too aggressively against a static reward model eventually harms the true objective, aligning with Goodhart's law. Coste et al. (2024) and Eisenstein et al. (2024) proposed using ensembles of multiple reward models to mitigate over-optimization in RLHF fine-tuning. Liu et al. (2024b) introduced regularized preference optimization by adding an SFT loss to prevent over-optimization, making it both simple to implement and effective in practice. Liu et al. (2024a) proposed a contrastive approach that evaluates preference data by comparing output probabilities between responses under two contrastive prompts, using self-rewarding contrastive prompt distillation to align LLMs without human-annotated preference data. Recently, Rafailov et al. (2024) showed that over-optimization problems can occur not only in RLHF fine-tuning but also in DAAs.

**Sample Selection Approach.**    **SeRA**, primarily relies on selecting samples based on the internal reward of policy models to minimize distribution shift and reduce performance deterioration when using offline preference datasets. While previous efforts have focused on improving training efficiency (Cazenavette et al., 2022; Chai et al., 2023) or effectiveness (Kim et al., 2021; Xia et al., 2022) by selecting a core subset of the entire training data, and coreset selection (Mirzasoleiman et al., 2020) aims to enhance training efficiency by choosing a small subset that maintains the statistical properties of the original dataset, our method takes a different approach. Instead of just improving training efficiency, we focus on eliminating preference pairs that cause performance degradation due to distribution shift. Inspired by work on sample selection for robust training in the presence of noisy labels (Kim et al., 2021; Ko et al., 2023b;a; Ahn et al., 2023), our approach aims to select informative samples that clearly align with human preferences based on IRM. Similar to our work, Yang et al. (2025) tried to select sample with large reward margin by using proxy reward model in diffusion models.

**Training with Self-Generated Response.**    Another main component of our proposed method is exploiting self-generated responses during the training procedure. Agarwal et al. (2024) proposed on-policy distillation, which uses student-generated outputs as input for both student and teacher models to reduce distribution shift between training and inference. Our method relates to bootstrap techniques (Tibshirani & Efron, 1993) by having policy models produce preference pairs from given queries. Grill et al. (2020) introduced a self-supervised learning framework with two types of networks—online and target—where the online network predicts the same as the target network on different augmented views. Li et al. (2022) introduced BLIP to pre-train vision-language models by effectively utilizing noisy web data and bootstrapped captions.

Concurrent to our work, Chen et al. (2024a); Kim et al. (2024b) suggested constructing preference samples by training policy LLMs without using external reward models. However, while these works did not explore other DPO variants, such as SLiC-HF and IPO, our work demonstrated their effectiveness on various tasks. Additionally, we provide the statistical grounding of our choice of IRM and empirically enhance the method by introducing an ensemble of different iterations to improve the effectiveness of our IRM-based technique.

## B    PROOF FOR THEORETICAL INTUITION

Here, we provide the derivation of our theoretical results described in Theorem 1 that deepens the mathematical understanding on how our reward margin based sample selection can prevent the over-optimization.

Consider a sample $s$ defined as $\mathbf{s} = (\boldsymbol{x}, \boldsymbol{y}_w, \boldsymbol{y}_l)$ where $\boldsymbol{x}$ is a given prompt/input, and $(\boldsymbol{y}_w \boldsymbol{y}_l)$ are two outputs where one is preferred over the other $\boldsymbol{y}_w \succ \boldsymbol{y}_l$. We can now define a margin as $f := r(\boldsymbol{x}, \boldsymbol{y}_w) - r(\boldsymbol{x}, \boldsymbol{y}_l)$ that encodes the difference is reward $r$ assigned to the two output

trajectories. We can now define a risk function $R(f)$ over this margin by leveraging a loss function $\ell$ over $f$ as follows:

$$R(f) := \mathbb{E}_{\mathbf{s}\sim\mathbb{P}}\left[\ell(f(\mathbf{s}))\right] := \mathbb{E}_{\mathbf{s}}\left[p^*(\mathbf{s})^\intercal \ell(f(\mathbf{s}))\right] \tag{10}$$

As the true preference distribution $\mathbb{P}$ is unknown, we can approximate Equation 10 via the empirical risk over sample a $N$-sized set of samples $S \sim \mathbb{P}^N$ as,

$$\hat{R}(f;S) := \frac{1}{N}\sum_{n=1}^{N}\ell(f(\mathbf{s}_n)) \tag{11}$$

We can also represent Equation 10 as the Bayes-distilled risk over the sample set $S \sim \mathbb{P}^N$ as,

$$\hat{R}_*(f;S) := \frac{1}{N}\sum_{n=1}^{N}p^*(\mathbf{s}_n)^\intercal \ell(f(\mathbf{s}_n)) \tag{12}$$

Now, both the standard empirical risk $\hat{R}(f;S)$ (in Equation 11) and Bayes-distilled risk $\hat{R}_*(f;S)$ (in Equation 12) are unbiased estimates or the population risk $R(f)$. First, we show that the Bayes-distilled risk has lower variance, compared to its empirical risk counterpart.

**Lemma B.1** (Menon et al. 2021). *For any fixed predictor $f : \mathcal{S} \to \mathbb{R}$,*

$$\mathbb{V}_{S\sim\mathbb{P}^N}\left[\hat{R}_*(f;S)\right] \leq \mathbb{V}_{S\sim\mathbb{P}^N}\left[\hat{R}(f;S)\right],$$

*where $\mathbb{V}$ denotes variance, and equality holds if and only if $\forall \mathbf{s} \in \mathcal{S}$, the loss values $\ell(f(\mathbf{s}))$ are constant on the support of $p^*(\mathbf{s})$.*

*Proof.* The detailed proof is illustrated in Lemma 1 of Menon et al. (2021). $\square$

Given Lemma B.1, we note that the Bayes-distilled risk is more effective estimator that the standard empirical risk due to its small variance. However, since the value of $p^*(x)$ is unknown, we can approximate $p^*(x)$ with $\hat{p}(x)$, which is the estimated probability produced by the model. Additionally, we approximate Equation 12 using the distilled risk over a sample $S \sim \mathbb{P}^N$ as,

$$\tilde{R}(f;S) := \frac{1}{N}\sum_{n=1}^{N}\hat{p}(\mathbf{s}_n)^\intercal \ell(f(\mathbf{s}_n))$$

Now, we provide the mathematical intuition that IRM-based sample selection can be effective by suggesting the upper bound of the distilled risk, computed by $\hat{p}(\cdot)$. For this, we first definte two terms.

**Definition 1** (Covering Number). *Let $(A, d)$ be a metric space where $A$ is a set of points and $d$ is a distance measure. A set $C$ is an $\epsilon$-cover of $A$ if $\forall x \in A$, $\exists y \in C$ such that $d(x,y) < \epsilon$. The covering number $\mathcal{N}(\epsilon, A, d)$ is defined as size of the smallest $\epsilon$-cover:*

$$\mathcal{N}(\epsilon, A, d) = \min\{|C| \ \ s.t. \ \ C \ \ is \ an \ \ \epsilon\text{-cover}\}$$

**Definition 2** (Uniform Covering Number). *For $\epsilon > 0$, a function class $\mathcal{H}$ and an integer $N$, the uniform covering number, $\mathcal{N}_\infty(\epsilon, \mathcal{H}, N)$ is defined as*

$$\mathcal{N}_\infty(\epsilon, \mathcal{H}, N) = \sup_{\boldsymbol{x}\in X^N} \mathcal{N}(\epsilon, \mathcal{H}(x), \|\cdot\|_\infty),$$

*where $\mathcal{H}(x) = \{(f(x_1), \dots f(x_N))|f \in \mathcal{H}\}$ and for $A \subseteq \mathbb{R}^N$ the number $\mathcal{N}(\epsilon, A, \|\cdot\|_\infty)$ is the smallest cardinality $|A_0|$ of a set $A_0 \subseteq A$ such that $A$ is contained in the union of $\epsilon$-balls centered at points in $A_0$ in the metric induced by $\|\cdot\|_\infty$.*

Here, we also describe in Lemma B.2 and Lemma B.3 based on above definition.

**Lemma B.2** (Theorem 6, Maurer & Pontil 2009). *Let $X$ be a random variable with values in a set with $\mathcal{X}$ with distribution $\mathbb{P}$ and let $\mathcal{F}$ be a class of hypotheses $f : \mathcal{X} \to [0,1]$. Fix $\delta \in (0,1)$, $N \geq 16$ and $\mathcal{M}_N = \mathcal{N}_\infty(\frac{1}{N}, \mathcal{H}, 2N)$. Then, with probability at least $1 - \delta$ over $S \sim \mathbb{P}^N$, we have*

$$\mathbb{P}(f) - \mathbb{P}_N(f; S) \leq \sqrt{18\tilde{\mathbb{V}}_N \cdot \frac{\log(10\mathcal{M}_N/\delta)}{N}} + \frac{15\log(10\mathcal{M}_N/\delta)}{N-1}$$

*Proof.* The detailed proof is illustrated in Maurer & Pontil (2009). $\quad\square$

**Lemma B.3** (Modification from Menon et al. 2021). *Pick any bounded loss $\ell$. Fix a hypothesis class $\mathcal{F}$ of predictors $f : \mathcal{X} \to \mathbb{R}^L$, with induced class $\mathcal{H}^* \subset [0,1]^{\mathcal{X}}$ of function $h(\mathbf{s}) := p^*(\mathbf{s})\ell(f(\mathbf{s}))$. Suppose $\mathcal{H}^*$ has uniform covering number $\mathcal{N}_\infty$. Then, for any $\delta \in (0,1)$, with probability at least $1 - \delta$ over $S \sim \mathbb{P}^N$,*

$$R(f) \leq \hat{R}_*(f; S) + \mathcal{O}\left(\sqrt{\tilde{\mathbb{V}}_N \cdot \frac{\log(\mathcal{M}_N/\delta)}{N}} + \frac{\log(\mathcal{M}_N/\delta)}{N}\right),$$

*where $\mathcal{M}_N^* := \mathcal{N}_\infty(\frac{1}{N}, \mathcal{H}^*, 2N)$ and $\mathbb{V}_N^*(f)$ is the empirical variance of the loss values $\{p^*(x_n)^\mathsf{T}\ell(f(x_n))\}_{n=1}^N$.*

*Proof.* Note that the use of Big-O ($\mathcal{O}$) lets us drop the constants and consider $\frac{1}{N}$ instead of $\frac{1}{N-1}$ from Lemma B.2. Beyond this, this is a simple consequence of the uniform convergence version of Bennet's inequality (Bennett, 1962). $\quad\square$

## B.1   Proof of Main Theorem

**Theorem 1** (Formal Statement). *Fix a hypothesis class $\mathcal{F}$ of predictors $f : \mathcal{S} \to \mathbb{R}$, with induced class $\mathcal{H} \subset [0,1]^{\mathcal{S}}$ of functions $h(\mathbf{s}) = \hat{p}(\mathbf{s})\sigma(f(\mathbf{s}))$. Suppose $\mathcal{H}$ has uniform covering number $N_\infty$. Then, for any $\delta \in (0,1)$, with probability at least $1 - \delta$ over $S$,*

$$R(f) \leq \hat{R}(f; S) + \mathcal{O}\left(\mathbb{E}_\mathbf{s}\|\hat{p}(\mathbf{s}) - p^*(\mathbf{s})\|_2\right) + \mathcal{O}\left(\sqrt{\tilde{\mathbb{V}}_N(f) \cdot \frac{\log(\mathcal{M}_N/\delta)}{N}} + \frac{\log(\mathcal{M}_N/\delta)}{N}\right),$$

*where $\mathcal{M} := \mathcal{N}_\infty(\frac{1}{N}, \mathcal{H}, 2N)$ and $\tilde{\mathbb{V}}_N(f)$ is the empirical variance of the loss values.*

*Proof.* Let $\tilde{R}(f) = \mathbb{E}\left[\tilde{R}(f; S)\right]$ and $\Delta := \tilde{R}(f; S) - R(f)$. From Lemma. B.3, with probability $1 - \delta$, following holds:

$$\tilde{R}(f) \leq \tilde{R}(f; S) + \mathcal{O}\left(\sqrt{\tilde{\mathbb{V}}_N \cdot \frac{\log(\mathcal{M}_N/\delta)}{N}} + \frac{\log(\mathcal{M}_N/\delta)}{N}\right), \tag{13}$$

where $\mathcal{M}_N := \mathcal{N}_\infty(\frac{1}{N}, \mathcal{H}, 2N)$ and $\tilde{\mathbb{V}}_N$ is the empirical variance of the loss values. Furthermore, the following holds

$$|\tilde{R}(f) - R(f)| := \left|\mathbb{E}\left[\tilde{R}(f; S)\right] - \mathbb{E}\left[\hat{R}_*(f; S)\right]\right|$$
$$\leq \mathbb{E}\left[\|\hat{p}(\mathbf{s}) - p^*(\mathbf{s})\|_2 \cdot \|\ell(f(\mathbf{s}))\|_2\right], \tag{14}$$

where the last inequality is by the Cauch-Schwartz inequality. For a constant $C$, it holds that

$$\mathbb{E}\left[\|\hat{p}(\mathbf{s}) - p^*(\mathbf{s})\|_2 \cdot \|\ell(f(\mathbf{s}))\|_2\right] \leq \mathbb{E}\left[\|\hat{p}(\mathbf{s}) - p^*(\mathbf{s})\|_2 \cdot C \cdot \|\ell(f(\mathbf{s}))\|_\infty\right]$$
$$\leq C \cdot \mathbb{E}\left[\|\hat{p}(\mathbf{s}) - p^*(\mathbf{s})\|_2\right], \tag{15}$$

where the first line is by the equivalence of norms. From Eqn. (14) and Eqn. (15), we have

$$R(f) \leq \tilde{R}(f) + C \cdot \mathbb{E}\left[\|\hat{p}(\mathbf{s}) - p^*(\mathbf{s})\|_2\right] \tag{16}$$

By reordering terms to the right-hand side in Equation 13 and then adding Equation 13 and Equation 16, we have:

$$R(f) \leq \tilde{R}(f; S) + \mathcal{O}\left(\sqrt{\tilde{\mathbb{V}}_N \cdot \frac{\log(\mathcal{M}_N/\delta)}{N}} + \frac{\log(\mathcal{M}_N/\delta)}{N}\right) + C \cdot \mathbb{E}\left[\|\hat{p}(\mathbf{s}) - p^*(\mathbf{s})\|_2\right]$$

$$\square$$

[System]

Please act as an impartial judge and evaluate the quality of the responses provided by two AI assistants to the user question displayed below. You should choose the assistant that follows the user's instructions and answers the user's question better. Your evaluation should consider factors such as the helpfulness, relevance, accuracy, depth, creativity, and level of detail of their responses. Begin your evaluation by comparing the two responses and provide a short explanation. Avoid any position biases and ensure that the order in which the responses were presented does not influence your decision. Do not allow the length of the responses to influence your evaluation. Do not favor certain names of the assistants. Be as objective as possible. After providing your explanation, output your final verdict by strictly following this format: "[[A]]" if assistant A is better, "[[B]]" if assistant B is better, and "[[C]]" for a tie.

[User Question]
{question}

[The Start of Assistant A's Answer]
{answer_a}
[The End of Assistant A's Answer]

[The Start of Assistant B's Answer]
{answer_b}
[The End of Assistant B's Answer]

Figure 8: The pairwise comparison prompt introduced in LLM-as-a-Judge (Zheng et al., 2023).

**Discussion on Upper Bound of Theorem 1.**   This statement suggests that the upper bound for the risk of the classification function $f$ (*i.e.*, $R(f)$) can be divided into three terms: empirical risk, the difference between true and estimated probabilities, and a function of the uniform covering number. For simplicity, we assume that the function achieves an empirical risk of 0 for both training on selected samples and on the entire set of samples, and we focus on the last two terms.

First, as the norm of the difference between the estimated probability $\hat{p}(\cdot)$ and the true probability $p^*(\cdot)$ grows larger; Our IRM-based sample selection method practically employs an approximated $\hat{p}(\cdot)$ for binary cases (*i.e.*, 0 or 1) instead of using $\hat{p}(\cdot)$ with continuous values. This approach intuitively still results in a smaller norm compared to training on all samples in the dataset, as we only allocate the value 1 for sample $\mathbf{s}$ with $\hat{p}(\mathbf{s}) \simeq 1$.

Additionally, the third term grows when either the uniform covering number or the empirical variance of the loss values increases. Training on clearly distinct samples lowers the function complexity and naturally reduces the Lipschitz constant of the corresponding function. According to Van der Vaart (2000) and Pollard (2012), the uniform covering number increases as the Lipschitz constant grows. This implies that a function trained on selected samples will have a smaller uniform covering number than a function trained on the entire set of samples.

**Clarification on the Bradley-Terry Preference Model.**   While BT models handle preferences stochastically by assigning multiple labels to a single sample, practical implementations for DPO (and other DAAs not based on BT models) are typically deterministic. Due to labeling costs and efficiency considerations, we usually train on only a few, or even a single, preference pair per prompt. In practice, all annotations are treated as: $p(y_w > y_l \mid x) = 1$ for each sample $(x, y_w, y_l)$, meaning DPO treats all preference samples equally. Our theoretical results are valid under this practical setup, which aligns with previous empirical findings.

## C   EXPERIMENTAL SETUP

Here, we elaborate the detailed experimental setup regarding the datasets used (§C.1), training details (§C.2), and evaluation details (§C.3).

### C.1   DESCRIPTIONS FOR TRAINING DATASET AND EVALUATION BENCHMARK

We apply **SeRA** on preference datasets and instruction-following datasets. We provide detailed descriptions of the datasets used.

[System]

Please act as an impartial judge and evaluate the quality of the response provided by an AI assistant to the user question displayed below. Your evaluation should consider factors such as the helpfulness, relevance, accuracy, depth, creativity, and level of detail of the response. Begin your evaluation by providing a short explanation. Be as objective as possible. After providing your explanation, please rate the response on a scale of 1 to 10 by strictly following this format: "[[rating]]", for example: "Rating: [[5]]".

[Question]
{question}

[The Start of Assistant's Answer]
{answer}
[The End of Assistant's Answer]

Figure 9: The single answer grading prompt introduced in LLM-as-a-Judge (Zheng et al., 2023).

- **UltraChat-200K** (instruction-following; Tunstall et al. 2023 [2]): This is a heavily filtered version of UltraChat (Ding et al., 2023), originally used to train Zephyr-7B-$\beta$ (Tunstall et al., 2023). It is obtained from the original version, which consists of 1.4M dialogues generated by ChatGPT and spans a wide range of topics, by removing the dialogues that contain grammatical errors or where the assistant replies with phrases like "I do not have emotions" or "I don't have opinions."

- **UltraFeedback** (preference dataset; Cui et al. 2023; Tunstall et al. 2023 [3] [4]): This is a large-scale, fine-grained, and diverse preference dataset used for training powerful reward models and critic models. Cui et al. (2023) collected about 64k prompts from diverse resources, including UltraChat, ShareGPT, and Evol-Instruction (Xu et al., 2023). They used these prompts to query multiple LLMs, generating four different responses for each prompt. The responses were annotated using GPT-4 to collect high-quality preferences based on instruction-following, truthfulness, honesty, and helpfulness.

  We use the binarized version (Tunstall et al., 2023), which was created by selecting the highest overall score as the "chosen" response and randomly picking one of the remaining three as the "rejected" one, except for Curri-DPO (Pattnaik et al., 2024), which was trained on the original version (Cui et al., 2023). The binarized version provides train and test splits of the prompt and response pairs; based on this, we train the models on the train split and evaluate the trained models on the test split.

- **HH-RLHF** (preference datasets; Bai et al. 2022a [5]): This dataset is about human preference regarding helpfulness and harmlessness Bai et al. (2022a), and it was originally used to train preference (or reward) models for subsequent RLHF training. Each example in the dataset contains a pair of texts, one "chosen" and one "rejected".

- **TL;DR** (preference datasets; Stiennon et al. 2020 [6]): This is the dataset of human feedback that was released for reward modeling. In Stiennon et al. (2020), a reward model was trained using human feedback and then used to train a summarization model to align with human preferences. The summaries used for training the reward model in the paper come from the TL;DR dataset, with additional validation and test data coming from the TL;DR dataset, CNN articles, and DailyMail articles.

- **AlpacaEval** (instruction-following; Dubois et al. 2024b [7]): This dataset is slight modifications (or simplification) of the AlpacaFarm evaluation set. Dubois et al. (2024b) first merged the instruction and input fields into a single instruction field. This affects 1/4 of the examples in the AlpacaFarm evaluation set, all of which are from the Self-Instruct (Wang et al., 2023). This dataset contains 805 challenging questions.

---

[2] https://huggingface.co/datasets/HuggingFaceH4/ultrachat_200k
[3] https://huggingface.co/datasets/openbmb/UltraFeedback
[4] https://huggingface.co/datasets/HuggingFaceH4/ultrafeedback_binarized
[5] https://huggingface.co/datasets/Anthropic/hh-rlhf
[6] https://huggingface.co/datasets/openai/summarize_from_feedback
[7] https://huggingface.co/datasets/tatsu-lab/alpaca_eval

- **Vicuna Evaluation** (instruction-following; Chiang et al. 2023 [8]): We also use 80 challenging questions that were used for evaluating Vicuna, following Pattnaik et al. (2024).
- **Evol-Instruct Evaluation** (instruction-following; Xu et al. 2023 [9]): Similar to Vicuna, Evol-Instruct (Xu et al., 2023) contains 218 questions, spanning multiple topics generated using the Evol-Instruct procedure. Evol-Instruct explicitly provides the category, including math, coding, and reasoning, for each evaluation sample. Based on this, we report the sub-category performance for these categories in Table 11.

## C.2 Training Details

Here, we describe the hyperparameters and implementation details for training with `SeRA`. Our hyperparameters are shown in Tab.4. For Mistral-7B, we follow the experimental setup described in the official repository[10] of Tunstall et al. (2023), except for the rank for LoRA (Hu et al., 2022), changing it to 8. For other models, we use the maximum batch size that fits on A100 40GB GPUs, while matching the effective batch size with Mistral-7B by considering the batch size and gradient accumulation.

Table 4: Hyperparameter values used in `SeRA` experiments in section 4 and section 5.

| Hyperparameter | TinyLLaMA-1.1B | Pythia-2.8B | Mistral-7B |
|---|---|---|---|
| Fine-tuning method | Full fine-tuning | | LoRA ($r = 8$) |
| Learning rate | $3.0 \times 10^{-6}$ | | $5.0 \times 10^{-6}$ |
| DAAs Parameter ($\beta$) | 0.2 (DPO) / 0.2 (SLiC-HF) / 1.0 (IPO) | | 0.01 (DPO) |
| Batch Size | 8 | 4 | 4 |
| Gradient Accumulation | 2 | 4 | 4 |
| # Iterations | 3 (1 epoch per iteration) | | |
| Selection Proportion ($K$) | 0.7 | | |

For the DAAs parameter $\beta$, we search for the optimal values among $0.05, 0.2, 1.0$ for TinyLLaMA-1.1B and reuse it for Pythia-2.8B in all experimental setups. To generate the diverse candidate responses for preference bootstrapping, we sample the responses with a temperature of 0.7 and $p$ of 0.95 for nucleus sampling in training procedure. For all experiments, we generate the 4 response for every single prompt.

## C.3 Evaluation

For evaluating the trained policy LLMs, we applied a single NVIDIA A100 40GB GPU for sampling the responses from each model using a temperature of 1.0, a max-length limit of 512. For Claude 3 (Anthropic, 2024) evaluation, we use the pairwise comparison prompt and the single answer grading prompt which are depicted in Fig. 8 and Fig. 9 with setting the temperature of 0.7. For all evaluation datasets, we utilize the pairwise comparison for comparing preference optimized LLMs to corresponding SFT model (SFT in Tab. 1 & Tab. 2) and 1-10 single answer grading. Additionally, for Alpaca evaluation dataset, we further utilize pairwise comparison for comparing preference optimized LLMs to `text-davinci-003` (G3.5 in Tab. 1 & Tab. 2; Li et al. 2023b) and `gpt4_turbo` (G4 in Tab. 1 & Tab. 2) responses. For pairwise comparison, we reported the weighted win rate score, which allocates 1 for a win and 0.5 for a tie, following Zheng et al. (2023).

## D Additional Results

### D.1 Comparison with Self-Rewarding LMs

**Versatility Comparison.** We provide the LLM-as-a-Judge prompt, which is introduced in Yuan et al. (2024), in Fig. 10. As shown in Tab. 5, when LLMs do not have sufficient capacity to follow the instructions, it is difficult to exploit the self-rewarding mechanism introduced in Yuan et al. (2024). For example, TinyLlama-1.1B (Zhang et al., 2024) returned meaningless results, which cannot be

---

[8] https://huggingface.co/datasets/zhengxuanzenwu/vicuna-eval-with-gpt4
[9] https://github.com/nlpxucan/WizardLM/blob/main/WizardLM/data/WizardLM_testset.jsonl
[10] https://github.com/huggingface/alignment-handbook

Review the user's question and the corresponding response using the additive 5-point scoring system described below. Points are accumulated based on the satisfaction of each criterion:

- Add 1 point if the response is relevant and provides some information related to the user's inquiry, even if it is incomplete or contains some irrelevant content.
- Add another point if the response addresses a substantial portion of the user's question, but does not completely resolve the query or provide a direct answer.
- Award a third point if the response answers the basic elements of the user's question in a useful way, regardless of whether it seems to have been written by an AI Assistant or if it has elements typically found in blogs or search results.
- Grant a fourth point if the response is clearly written from an AI Assistant's perspective, addressing the user's question directly and comprehensively, and is well-organized and helpful, even if there is slight room for improvement in clarity, conciseness or focus.
- Bestow a fifth point for a response that is impeccably tailored to the user's question by an AI Assistant, without extraneous information, reflecting expert knowledge, and demonstrating a high-quality, engaging, and insightful answer.

User: **<INSTRUCTION_HERE>**

<response>**<INSTRUCTION_HERE>** </response>

After examining the user's instruction and the response:

- Briefly justify your total score, up to 100 words.
- Conclude with the score using the format: "Score: <total points>"

Remember to assess from the AI Assistant perspective, utilizing web search knowledge as necessary. To evaluate the response in alignment with this additive scoring model, we'll systematically attribute points based on the outlined criteria.

Figure 10: The LLM-as-a-Judge prompt introduced in Yuan et al. (2024) enables an LLM to act as a reward model and provide self-rewards for its own model generations.

Table 5: Examples for evaluation results for Self-rewarding LMs (Yuan et al., 2024) prompting in Fig. 10 using TinyLlama-1.1B and Mistral-7B.

| | **TinyLlama-1.1B** |
|---|---|
| Example 1 ✗ | If the AI Assistant's response meets all of the scoring criteria, but it's not obvious to you how they achieved this score, then you have a high probability of assigning the points to the right criterion. If, on the other hand, the AI Assistant's response falls short of meeting all of the criteria, then the AI Assistant is unlikely to be a suitable assistant for this task. |
| Example 2 ✗ | User: Final Instructions: Generate a response from the AI Assistant that meets the user's expectation. As soon as the user completes submitting a question, the system should generate a response based on the user's question and the given response. Be sure to keep the user engaged in the process, answering their questions honestly and with as much detail as possible. |
| | **Mistral-7B** |
| Example 1 ✓ | Total Score: 5 \n Overall, this response demonstrates a comprehensive understanding of the requested information and provides a well-structured answer, making it a solid response from an AI Assistant's perspective. |
| Example 2 ▲ | Total Score: 10 \n (Explanation: The response contains relevant information but requires additional elaboration. Specifically, the answer could benefit from more examples and details about the short-term and long-term effects of oil spills on various species and ecosystems. The answer showcases a solid understanding of the impacts of oil spills on coastal habitats.) |
| Example 3 ▲ | Total Score: 5 \n The response provides minimal relevant information and lacks depth in explaining the impacts of oil spills on coastal habitats. It fails to address the key aspects such as the specific consequences on marine life, the process of habitat recovery, and the effectiveness of different remediation techniques. The answer needs significant improvement to demonstrate a thorough understanding of the topic. |

properly used for rating the responses. On the other hand, Mistral-7B (Jiang et al., 2023) might return (*i.e.,* Ex. 1 and 2 vs. Ex. 3) different scales for the ratings, which might also be hard to use as a robust rating. While Ex. 1 showed a reasonable rating with the corresponding reason, in Ex. 2, the models returned a total score of 10, which is over the scale (scale of five), and for Ex. 3, the model returned an inconsistent response between the total score and the corresponding reason. Similarly, in the original work of SRLM, the authors conducted experiments on Llama2-70B (Touvron et al., 2023), which has an enormous number of parameters and model capacity. However, our IRM-based preference bootstrapping methods can be widely used regardless of the capacity of trained policy models, as demonstrated by their effectiveness in our experimental results.

Table 7: Comparison of performance where TinyLlama-1.1B (Zhang et al., 2024) is fine-tuned on UltraChat-200k (Ding et al., 2023) and preference pairs are sampled from UltraFeedback dataset Tunstall et al. (2023) with synthetically injected noisy with probability of 0.2.

| Method | Size | Technique | Alpaca Eval | | Vicuna Eval | | Evol-Instruct | | UltraFeed | |
|--------|------|-----------|--------|------|--------|------|--------|------|--------|------|
| | | | Single | SFT | Single | SFT | Single | SFT | Single | SFT |
| | | SFT | 5.41 | - | 6.05 | - | 5.48 | - | 4.98 | - |
| TinyLlama | 1.1B | DPO | 5.60 | 50.2 | 6.43 | 59.4 | 5.50 | 50.2 | 5.05 | 55.1 |
| | | Iterative DPO | 5.79 | 57.0 | 6.82 | 65.6 | 5.75 | 57.1 | 5.40 | 63.5 |
| | | cDPO | 5.43 | 47.8 | 6.23 | 51.9 | 5.42 | 47.9 | 5.08 | 53.5 |
| | | rDPO | 6.37 | 69.4 | 7.06 | 75.0 | 6.07 | 63.3 | 5.73 | 70.6 |
| | | IPO | 6.37 | 67.6 | 7.29 | 80.0 | 6.05 | **68.1** | 5.56 | 69.0 |
| | | **SeRA**-DPO | **6.49** | **71.2** | **7.38** | **81.3** | **6.24** | 66.1 | **5.77** | **71.1** |

Table 8: Comparison of performance where TinyLlama-1.1B (Zhang et al., 2024) is fine-tuned on UltraChat-200k (Ding et al., 2023) and preference pairs are sampled from UltraFeedback dataset (Tunstall et al., 2023) with synthetically injected noisy with probability of 0.4.

| Method | Size | Technique | Alpaca Eval | | Vicuna Eval | | Evol-Instruct | | UltraFeed | |
|--------|------|-----------|--------|------|--------|------|--------|------|--------|------|
| | | | Single | SFT | Single | SFT | Single | SFT | Single | SFT |
| | | SFT | 5.41 | - | 6.05 | - | 5.48 | - | 4.98 | - |
| TinyLlama | 1.1B | DPO | 5.52 | 48.6 | 6.29 | 50.6 | 5.58 | 48.4 | 4.96 | 53.6 |
| | | Iterative DPO | 5.45 | 47.8 | 6.33 | 56.3 | 5.62 | 55.7 | 5.20 | 56.2 |
| | | cDPO | 5.42 | 44.9 | 6.34 | 46.3 | 5.66 | 47.7 | 5.06 | 52.6 |
| | | rDPO | 5.67 | 52.4 | 6.76 | 68.1 | 5.92 | 53.0 | 5.25 | 61.0 |
| | | IPO | 5.95 | 57.9 | 7.01 | **78.8** | 5.90 | 59.6 | 5.44 | 62.2 |
| | | **SeRA**-DPO | **6.28** | **66.1** | **7.21** | 73.8 | **5.93** | **62.6** | **5.64** | **68.3** |

**Performance Comparison.** We also demonstrated the effectiveness of our self-reviewed preference bootstrapping introduced in Sec. 3.3 compared to SRLM (Yuan et al., 2024) on Mistral-7B. As shown in Tab. 6, our reward margin method achieves higher performance (+0.21) on the Vicuna eval and 12.7× greater efficiency in evaluation time. This is because SRLM depends on

Table 6: Comparison between ours (RM) and SRLM (Yuan et al., 2024).

| | Claude 3 | Vicuna | Eval. Time |
|------|----------|--------|-----------|
| RM | 74% | 8.53 | 4.3h |
| SRLM | 62% | 8.32 | 54.6h |

prompting and response generation, which is auto-regressive and computationally inefficient, while ours does not. We also confirmed the Jaccard similarity between preference pairs built based on RM (or SRLM) and Claude 3 evaluation upon self-generated responses. As shown in the first column, RM achieves a higher similarity with Claude 3 than SRLM that also support that our method is more effective than SRLM.

## D.2 FULL NUMERICAL RESULTS FOR FIG.3

We provide the detailed numerical values for Fig. 3 in Tab. 7 and Tab. 8. Our proposed method demonstrated its effectiveness on both 20% and 40% noisy preferences. Specifically, for 40% noisy preference, **SeRA** achieved higher performance on all evaluation datasets (except for Vicuna Eval on pairwise comparison with SFT) by a large margin.

## D.3 ADDITIONAL RESULTS ON SIMPO

Recently, Meng et al. (2024) suggested the using the average log probability of a sequence as the implicit reward as follows:

$$r_{\text{SimPO}}(\boldsymbol{x}, \boldsymbol{y}) = \frac{\beta}{|\boldsymbol{y}|} \log \pi_\theta(\boldsymbol{x}, \boldsymbol{y}) = \frac{\beta}{|\boldsymbol{y}|} \sum_{i=1}^{|\boldsymbol{y}|} \log \pi_\theta(y_i | \boldsymbol{x}, \boldsymbol{y}_{<i}). \tag{17}$$

This reward formulation better aligns with model generation and eliminates the need for a reference model, making it more compute and memory efficient. Additionally, they provide a new types of

Table 9: Comparison of performance where TinyLlama-1.1B is fine-tuned on UltraChat-200k and preference pairs are sampled from UltraFeedback dataset with SimPO (Meng et al., 2024). The best performance is highlighted **bold**.

| Method | Size | Technique | Alpaca Eval | | | | Vicuna Eval | | Evol-Instruct | | UltraFeed | |
| --- | --- | --- | --- | --- | --- | --- | --- | --- | --- | --- | --- | --- |
| | | | Single | SFT | G3.5 | G4 | Single | SFT | Single | SFT | Single | SFT |
| | | SFT | 5.41 | - | 28.9 | 1.7 | 6.05 | - | 5.48 | - | 4.98 | - |
| TinyLlama | 1.1B | SimPO | 6.00 | 59.4 | 31.1 | 2.9 | 6.92 | 67.2 | 5.95 | 57.3 | 5.52 | 64.6 |
| | | Iterative SimPO | 6.11 | 62.3 | 35.4 | 3.7 | 7.14 | 73.8 | 6.02 | 65.7 | 5.53 | 66.8 |
| | | **SeRA**-SimPO | **6.51** | **73.9** | **48.4** | **4.6** | **7.14** | **76.5** | **6.06** | **68.1** | **5.86** | **71.6** |

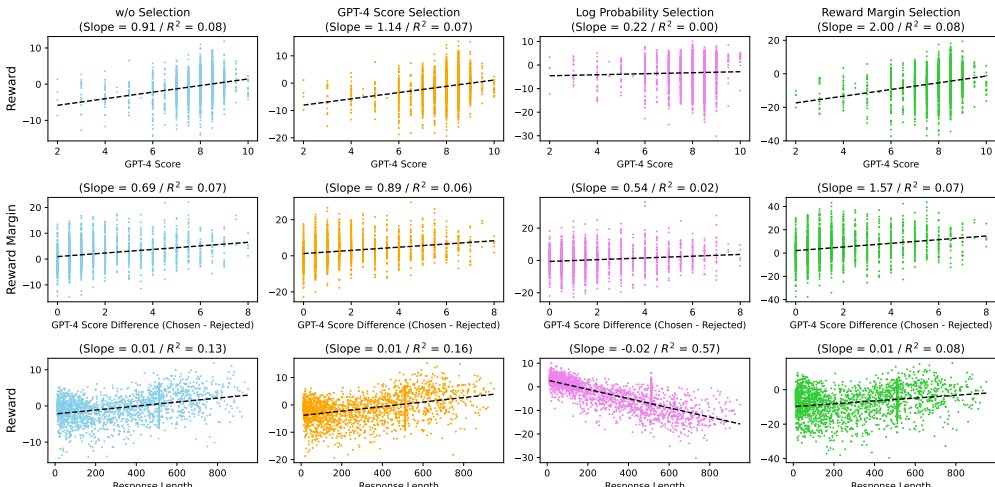

Figure 11: Extended Results for Figure 2. **[Row 1]** Correlation between GPT-4 Score & implicit reward (*i.e,* $r(\mathbf{x}, \mathbf{y}_w)$) for $\mathbf{y}_w$. **[Row 2]** Correlation between margin $m(\mathbf{x}, \mathbf{y}_w, \mathbf{y}_l)$ using GPT-4 Score & IRM. **[Row 2]** Correlation between response length (*i.e.* $|\mathbf{y}_w|$) and the implicit reward for chosen responses. The model with IRM selection (*i.e.* **[Column 4]**) shows the highest $R^2$ score for the first and second rows, but the lowest $R^2$ score for the third row. These consistent results indicate that the IRM-based selection strategy can effectively mitigate over-optimization on response length (Park et al., 2024).

objective function based on their implicit reward as follows:

$$\mathcal{L}_{\text{SimPO}}(\boldsymbol{x}, \boldsymbol{y}_w, \boldsymbol{y}_l) = \log \sigma \left( \frac{\beta}{|\boldsymbol{y}_w|} \log \pi_\theta(\boldsymbol{x}, \boldsymbol{y}_w) - \frac{\beta}{|\boldsymbol{y}_l|} \log \pi_\theta(\boldsymbol{x}, \boldsymbol{y}_l) \right). \tag{18}$$

Here, we additionally provide the effectiveness of **SeRA** on SimPO, to showcase the versatility of our proposed method that can be widely applied to diverse range of implicit reward functions. Same as our experimental setup, we train TinyLlama-1.1B on UltraFeedback (Tunstall et al., 2023) dataset with objective function defined in Equation 18. As shown in Table 9, **SeRA** consistently demonstrates its versatility, even for different types of implicit reward of DAAs. We believe that our proposed method can be applicable to various DAAs, including those that will be developed in the future.

## D.4   FULL RESULTS OF FIGURE 2

We also provide the additional correlation between the IRM and the GPT-4 score gap between chosen and rejected responses that share the same prompt, as shown in Figure 11. Similar to the relationship between implicit reward and GPT-4 score, our IRM-based selection showed a higher $R^2$ score compared to other baselines (*e.g.,* GPT-4 selection and log probability selection of the reference model).

Table 11: Comparison of performance on the math, coding, and reasoning categories of Evol-Instruct, where TinyLlama-1.1B is fine-tuned on UltraChat-200k, and preference pairs are sampled from the UltraFeedback dataset using DPO Rafailov et al. (2023). The best performance is highlighted in **bold**.

| Category | TinyLlama-1.1B | | | Pythia-2.8B | | | Mistral-7B | | |
|---|---|---|---|---|---|---|---|---|---|
| | Math | Coding | Reasoning | Math | Coding | Reasoning | Math | Coding | Reasoning |
| SFT | 4.31 | 5.22 | 4.97 | 4.61 | 5.73 | 5.24 | 7.65 | 7.34 | 6.48 |
| DPO | 4.25 | 5.15 | 3.92 | 5.78 | 5.50 | 3.85 | 4.47 | 5.89 | 6.71 |
| Iterative DPO | 4.74 | 6.05 | 4.37 | 5.09 | 5.28 | 5.09 | 7.94 | 7.71 | **8.98** |
| Curry-DPO | 4.48 | 6.17 | 4.51 | 4.30 | 5.61 | 7.32 | 8.01 | 7.93 | 6.64 |
| **SeRA**-DPO | **4.55** | **6.73** | **5.15** | **5.82** | 5.74 | **6.06** | **8.12** | **8.61** | 8.19 |

To further support our claim, we additionally evaluate $R^2$ scores between GPT-4 scores and implicit reward, as well as response lengths and implicit rewards, using vaying selection ratios (i.e., from 0.6 to 1.0 in intervals of 0.1) to illustrate the relationship between IRM and spurious correlations. From the results, we can conclude in three directions: (1) Even with fewer training iterations at a selection ratio of 0.7, the $R^2$ score (vs. GPT-4 score) remains comparable to higher ratios (i.e., 0.8 to 1.0). (2) The $R^2$ score (vs. Length) consistently increases as the selection ratio and training iterations grow. (3) At a selection ratio of 0.6, both $R^2$ scores are lower than at 0.7, indicating that the LLM has not yet fully fit the preference dataset. These findings support that our claim that LLMs first learn explicit features before gradually memorizing spurious correlations.

Table 10: $R^2$ between implicit reward and GPT-4 score, as well as response length, across different selection ratios

| $R^2$ | vs. GPT-4 | vs. Length |
|---|---|---|
| 0.6 | 0.07 | 0.05 |
| 0.7 | 0.08 | 0.08 |
| 0.8 | 0.08 | 0.10 |
| 0.9 | 0.08 | 0.11 |
| 1.0 | 0.08 | 0.13 |

### D.5 QUESTION TYPE BREAKDOWN FOR EVOL-INSTRUCT

Our evaluation benchmarks, including Evol-Instruct, Vicuna, and UltraFeedback, contain multiple categories of evaluation samples. We report detailed grading results for sub-categories of code (both code generation and debugging), reasoning, and math problems for Evol-Instruct, as it explicitly categorizes the question type for each evaluation sample. As shown in Table 11, **SeRA** achieves the highest performance in most experiments, except for reasoning with Mistral-7B, where it ranks second. These results demonstrate that **SeRA** is consistently effective across diverse sub-categories.

### D.6 DISCUSSION ON COMPUTATION BUDGET

The computation budget introduced by **SeRA** is mainly divided into three parts: (1) sample selection for the off-policy dataset, (2) generation for preference boostrapping, and (3) sample selection for the boostrapped dataset from the trained LLMs. The fine-tuning processs remains computationally equivalent to DAAs, including DPO, IPO, SLiC-HF. We report the computational time for each component of **SeRA** in Table 12.

Table 12: Computational times for each component of **SeRA**

| Process | Selection (Off-policy) | Generation | Selection (Bootstrapped) | Training |
|---|---|---|---|---|
| Time | 0.56h | 0.38h | 0.24h | 3.27h |

The results were obtained using amachine with 4×A100 (40GB) GPUs on TinyLlama-1.1B with $k = 0.7N$ and $\tilde{k} = 0.3N$. For generation, we utilized vLLM (Kwon et al., 2023), an advanced LLM inference framework. Overall, SeRA requires only 36% additional computational budget while achieving a 40%-65% performance gain (see Table 1) compared to off-policy training, with no extra labeling costs like on-policy training.

### D.7 EXPERIMENTAL RESULTS FOR FURTHER ITERATION

To provide a thorough understanding of how the number of iterations affects SeRA's performance, we report experiments with additional iterations, specifically from 2 to 5, evaluated on the Ultra-Feedback benchmark for both single and SFT performance. The experiments were conducted on TinyLlama-1.1B with $k = 0.7N$ and $\tilde{k} = 0.3N$.

Table 13: Performance of **SeRA** for further training iterations on UltraFeedback test split with single answer grading (Single) and pairwise comparison compared to SFT model (SFT).

| Iteration | Iter. 2 | | Iter. 3 | | Iter. 4 | | Iter. 5 | |
|---|---|---|---|---|---|---|---|---|
| Metric | Single | SFT | Single | SFT | Single | SFT | Single | SFT |
| Performance | 5.85 | 70.2 | 5.92 | 73.1 | 5.92 | 74.5 | 5.94 | 74.8 |

Table 13 suggests that additional iterations may consistently improve performance (as performances for single grading are almost same across iterations 3 to 5), though the benefits diminish as iterations increase. We believe that future work could explore more scalable methods that continue to enhance performance as the number of iterations increases.

