# OpenReview forum: "SeRA: Self-Reviewing and Alignment of LLMs using Implicit Reward Margins"
_ICLR.cc/2025/Conference — ICLR 2025 Poster_

### Official Review · Reviewer_AqA6 · 2024-10-30

**Soundness:** 3
**Presentation:** 3
**Contribution:** 3
**Rating:** 6
**Confidence:** 2

**Summary:**

Direct alignment algorithms (DAAs) like direct preference optimization (DPO) face challenges with static off-policy data, leading to spurious correlations and overfitting. In this paper, the authors propose a novel method called SeRA that addresses the mentioned issue by using sample selection and preference bootstrapping to improve alignment. The authors use extensive experiments to show SeRA enhances DAA performance on the Ultrafeedback dataset.

**Strengths:**

This paper provides a clear motivation for the authors' proposed method from both theoretical and empirical perspectives. The experimental results show that their proposed method has a significant improvement on the alignment task given the Ultrafeedback dataset and the authors perform extensive empirical analysis to explain the mechanism behind the proposed method.

**Weaknesses:**

1. The paper lacks more detailed discussions on the literature analyzing the overoptimization issue for LLM alignment and RLHF with self-generated data. For example, the authors could discuss the literature [1]-[5] in this paper.


2. Compared with other papers that study LLM alignments (e.g. [5]. [6]), this paper lacks empirical results on the academic benchmarks in coding, reasoning, and math. The authors should evaluate their proposed methods on these academic benchmarks.

3. It is a little confusing that the authors leave Remark 1 in Line 254 without an in-depth explanation of the derived bound. The derived bound seems to be adapted from the common analysis of empirical risk minimization. If there are novel analytical techniques for the current problem (analyze the performance of the proposed method), the authors should highlight the novel part and show that the vanilla DPO fails to have such a bound.

4. Minor issues or suggestions:
(1) In the comment on Line 14 of Algorithm 1, you should better mention the 'implicit reward' instead of 'reward'.
(2) More descriptions of benchmarks should be given. (even in the appendix)


[1] Coste, Thomas, et al. "Reward Model Ensembles Help Mitigate Overoptimization." The Twelfth International Conference on Learning Representations.

[2] Liu, Zhihan, et al. "Provably mitigating overoptimization in rlhf: Your sft loss is implicitly an adversarial regularizer." arXiv preprint arXiv:2405.16436 (2024).

[3] Eisenstein, Jacob, et al. "Helping or herding? reward model ensembles mitigate but do not eliminate reward hacking." arXiv preprint arXiv:2312.09244 (2023).

[4] Liu, Aiwei, et al. "Direct large language model alignment through self-rewarding contrastive prompt distillation." arXiv preprint arXiv:2402.11907 (2024).

[5] Wu, Yue, et al. "Self-play preference optimization for language model alignment." arXiv preprint arXiv:2405.00675 (2024).

[6] Chen, Zixiang, et al. "Self-play fine-tuning converts weak language models to strong language models." arXiv preprint arXiv:2401.01335 (2024).

**Questions:**

1. In Line 246, the authors claim that the inherent problem of DPO is that DPO treats all preference samples equally, even when the true preferences are not correct. In the BT preference model, the annotation is stochastic instead of deterministic, which means that this phenomenon should not be an issue from the theoretical perspective. Could the authors give a more detailed explanation?

2. The proposed method requires sampling multiple times for each prompt in the dataset. I wonder if the baseline methods and the proposed method have the same computation budget.

3. In this paper, the authors consider introducing a filtration on self-generated responses to get a preference pair for each prompt. However, I am curious about what would happen if the authors use all the self-generated responses (e.g. N responses) in pairs to create N(N-1)/2 preference pair. Such an idea is used in SPPO ([1]), hence a comparison between the proposed method in this paper and SPPO may be important.

4. Since the authors use the Ultrafeedback dataset for training, I am curious about the meaning of the 'Ultrafeedback' benchmark. Is this the test split of the Ultrafeedback?


[1] Wu, Yue, et al. "Self-play preference optimization for language model alignment." arXiv preprint arXiv:2405.00675 (2024).

---

> ### Author Response · Authors · 2024-11-20
> **Response to Reviewer AqA6 (1/3)**
>
> We are very grateful for your constructive comments. We have expressed your concerns as we understood them and provided answers as follows. If there is any concern that we may have misunderstood or if you have additional questions, we look forward to addressing them through further comments from you.
>
> **Q1. Additional literature review**
>
> > A1. Thank you for pointing out the missing references [1-6]. We have expanded the literature review to address the over-optimization issue in LLM alignment. Regarding self-generated data, we have already included relevant discussions in the appendix; however, we have expanded on this further based on your suggestions. All updates can be found in the latest version of our paper.
>
> **Q2. Further extension to domains including coding, reasoning, and math**
>
> > A2. Our evaluation benchmarks, including Evol-Instruction, Vicuna, and UltraFeedback, already contain coding, reasoning, and math tasks. We have reported detailed grading results for the sub-categories of code (both code generation and code debugging), reasoning, and math problems using Evol-Instruct, which explicitly categorizes each evaluation sample. The results below are for TinyLlama-1.1B, Pythia-2.8B, and Mistral-7B:
>
> | Benchmark     |          | TinyLlama |           |          |  Pythia  |           |          |  Mistral |           |
> |---------------|:--------:|:---------:|:---------:|:--------:|:--------:|:---------:|:--------:|:--------:|:---------:|
> | Category      |   Math   |   Coding  | Reasoning |   Math   |  Coding  | Reasoning |   Math   |  Coding  | Reasoning |
> | SFT           |   4.31   |    5.22   |    4.97   |   4.61   |   5.73   |    5.24   |   7.65   |   7.34   |    6.48   |
> | DPO           |   4.25   |    5.15   |    3.92   |   5.78   |   5.50   |    3.85   |   4.47   |   5.89   |    6.71   |
> | Iterative DPO |   4.74   |    6.05   |    4.37   |   5.09   |   5.28   |    5.09   |   7.94   |   7.71   |  **8.98** |
> | Curry-DPO     |   4.48   |    6.17   |    4.51   |   4.30   |   5.61   |    7.32   |   8.01   |   7.93   |    6.64   |
> | SeRA-DPO      | **4.55** |  **6.73** |  **5.15** | **5.82** | **5.74** |  **6.06** | **8.12** | **8.61** |    8.19   |
>
> > As shown in the table, SeRA achieves the highest performance in most categories, except for reasoning with Mistral, where it ranks second. These results demonstrate that SeRA is also effective across coding, reasoning, and math problems.

---

> ### Author Response · Authors · 2024-11-20
> **Response to Reviewer AqA6 (2/3)**
>
> **Q3. Further explanation for Remark 1 (or Theorem 1)**
>
> > A3. To address the reviewer's concerns, we provide clarifications in two directions:
>
> > 1. **Explanation of Remark 1 (and Theorem 1)**:  The purpose of Remark 1 (or Theorem 1) is to provide mathematical support for the empirical phenomena observed in **Fig. 2**, which motivated our use of the implicit reward margin (IRM). Theorem 1 indicates that the upper bound of the risk \( f(x, y_w, y_l) \) is reduced when training on samples with clear preferences, as opposed to using a dataset that includes ambiguous samples. This helps prevent memorization of spurious features and reduces the gap between predicted and true probabilities, as illustrated by the second term on the RHS of the formula in Remark 1. These theoretical results align with our observations in **Fig. 2**.
>
> > 2. **Clarification on the BT Preference Model**:  While BT models handle preferences stochastically by assigning multiple labels to a single sample, practical implementations for DPO (and other DAAs not based on BT models) are typically deterministic. Due to labeling costs and efficiency considerations, we usually train on only a few, or even a single, preference pair per prompt. In practice, all annotations are treated as:  $p(y_w > y_l \mid x) = 1$ for each sample $(x, y_w, y_l)$, meaning DPO treats all preference samples equally. Our theoretical results are valid under this practical setup, which aligns with previous empirical findings.
>
> > To further support our theoretical results, we evaluated the $R^2$ scores between GPT-4 scores vs. implicit rewards, as well as response length vs. implicit rewards, using varying selection ratios (0.6 to 1.0 in intervals of 0.1) to illustrate the relationship between IRM and spurious correlations.
>
> | R^2 (Implicit Reward) |  0.6 |  0.7 |  0.8 |  0.9 |  1.0 |
> |-----------------------|:----:|:----:|:----:|:----:|:----:|
> | vs. GPT-4             | 0.07 | 0.08 | 0.08 | 0.08 | 0.08 |
> | vs. Length            | 0.05 | 0.08 | 0.10 | 0.11 | 0.13 |
>
> > From these results, we draw three key conclusions:
>
> > 1. Even with fewer training iterations at a selection ratio of 0.7, the $R^2$ score (vs. GPT-4) remains comparable to higher ratios (0.8 to 1.0).
> > 2. The $R^2$ score (vs. Length) consistently increases as the selection ratio and training iterations grow.
> > 3. At a selection ratio of 0.6, both $R^2$ scores are lower than at 0.7, indicating that the LLM has not yet fully fit the dataset.
>
> > These findings support our claim that LLMs first learn explicit features (i.e., true preferences) before gradually memorizing spurious correlations (e.g., response length). Our use of IRM helps the model focus on true preferences while minimizing spurious correlations, supporting both mathematically and empirically, thereby justifying its use as a core component of our proposed method.
>
> **Q4. Minor suggestions regarding terminology clarification and benchmark details**
>
> > A4. We modified "reward" to "implicit reward" in Algorithm 1 and added more descriptions for the benchmark in Appendix C.1, highlighted in magenta. Thank you for your suggestions.
>
>
> **Q5. Computation budget for SeRA**
>
> > A5. The cost overhead introduced by SeRA is mainly divided into three parts: (1) sample selection for the off-policy dataset, (2) generation for preference bootstrapping, and (3) sample selection for the bootstrapped dataset from the trained LLM. The fine-tuning process remains computationally equivalent to DAAs. Below, we report the computational time for each component of SeRA:
>
> | Process | Selection (Off-policy) | Generation | Selection (Bootstrapped) | Training |
> |:-------:|:----------------------:|:----------:|:-----------------------:|:--------:|
> |   Time  |          0.56h         |    0.38h   |          0.24h          |   3.27h  |
>
> > Results were obtained using a machine with 4xA100 (40G) GPUs on TinyLlama-1.1B with 𝑘 = 0.7N and $\tilde{k}$ = 0.3N. For generation, we utilized vLLM [7], an advanced LLM inference framework. Overall, SeRA requires only 36% additional computational budget while achieving a 40%-65% performance gain (see Table 1 in the main manuscript) compared to off-policy training, with no extra labeling costs like on-policy training.

---

> ### Author Response · Authors · 2024-11-20
> **Response to Reviewer AqA6 (3/3)**
>
> **Q6. Discussion on dataset construction proposed in SPPO**
>
> > A6. We would like to clarify that our experiments have already shown that increasing the number of self-generated samples can be beneficial (e.g., best-SeRA with $k=0.7N$ and $\tilde{k}=2N$, as presented in **Table 1** and **Fig. 7**). This aligns with the reviewer's suggestion of using all possible preference pairs in the training dataset.
> However, as observed in Fig. 7, performance consistently improves up to 2N pairs, but starts to degrade beyond that, particularly at 3N where the number of responses is 3 (i.e., $R=3$ where $R$ is number of generations per single prompt) for SPPO dataset construction. This suggests that using all possible preference pairs is not always advantageous.
>
> > Additionally, while SPPO benefits from using more preference pairs, it relies on external reward models and introduces a different objective function, which diverges from our setup. The use of additional pairs may be effective in their framework, but may not be feasible in scenarios with limited budget constraints, which our method specifically addresses.
>
> **Q7. Clarification for UltraFeedback benchmark**
>
> > A7. Yes, that’s correct. As mentioned in the main text (line 358) and the appendix (line 1074), we used the binarized UltraFeedback dataset, which includes a test split for evaluation.
>
> ## References
> [1] Coste et al., “Reward Model Ensembles Help Mitigate Overoptimization.” ICLR. 2024 \
> [2] Liu et al., “Provably Mitigating Overoptimization in RLHF: Your SFT Loss is Implicitly an Adversarial Regularizer”. arXiv preprint. 2024 \
> [3] Eisenstein et al., “Helping or Herding? Reward Model Ensembles Mitigate but do not Eliminate Reward Hacking.” CoLM. 2024 \
> [4] Liu et al., Direct Large Language Model Alignment Through Self-Rewarding Contrastive Prompt Distillation.” arXiv preprint. 2024 \
> [5] Gao et al., “Scaling Laws for Reward Model Overoptimization.” ICML. 2023 \
> [6] Rafailov et al., “Scaling Laws for Reward Model Overoptimization in Direct Alignment Algorithms.” NeurIPS. 2024 \
> [7] Kwon et al., “Efficient Memory Management for Large Language Model Serving with PagedAttention”. SOSP. 2023

---

> ### Author Response · Authors · 2024-11-25
> **Gentle Reminder: Response Review for Reviewer AqA6**
>
> Dear Reviewer AqA6,
>
> In our response, we have addressed your questions by providing:
>
> - **An additional literature review on over-optimization for LLM alignment**,
> - **Additional results for specific domains, including coding, reasoning, and math**,
> - **An expanded description of Remark 1 and further experimental results to support our claims**,
> - **An additional investigation into computational costs**,
> - **A discussion of the dataset construction introduced in SPPO**, and
> - **A clarification of the evaluation benchmark**.
>
> We hope our response provides additional clarity regarding your questions. We would appreciate it if you could confirm that you have read our response and share any further questions you might have before the discussion period ends.
>
> Thank you for your time.
>
> Sincerely, \
> The Authors

---

> ### Author Response · Authors · 2024-11-28
> **Kindly Following Up: Response Review for Reviewer AqA6**
>
> *Dear Reviewer AqA6,*
>
> We hope our response and revised manuscript provided additional information regarding your questions. We would appreciate it if you could acknowledge that you have read our response and share any further questions or concerns before the discussion period ends.
>
> If there are specific points you’d like us to clarify, please let us know, and we would be happy to elaborate to address your concerns.
>
> If there are no further questions, we hope we have satisfactorily addressed your concerns and would be delighted if you might consider reflecting this in your re-evaluation.
>
> Sincerely, \
> The Authors

---

> ### Comment · Reviewer_AqA6 · 2024-12-03
> **Reply to the Authors**
>
> Thank the authors for the replies and I will keep my positive scores.

---

### Official Review · Reviewer_pPvp · 2024-11-02

**Soundness:** 3
**Presentation:** 3
**Contribution:** 3
**Rating:** 6
**Confidence:** 3

**Summary:**

The paper presents a new algorithm called Self-Reviewing and Alignment (SeRA) for direct aliment algorithms, which can be adapted to different preference optimization methods such as DPO, SLiC, and IPO. The two components of SeRA are sample selection and preference bootstrapping. Instead of using human feedback, AI judgment, or an external reward model to label the preference data, SeRA uses an implicit reward model for the sample selection and data bootstrapping. The paper provides a theoretical guarantee for DPO. Also, the paper presents extensive empirical studies across different datasets, model architectures, and alignment frameworks to show the effectiveness of SeRA.

**Strengths:**

* The paper presents both theoretical and empirical validation, covering a variety of models, datasets, and SAA methods. Furthermore, the component analysis is thorough, evaluating most elements of the algorithm for a more nuanced understanding.
*  The IRM method enables training SAA iteratively without relying on external reward signals. This innovation simplifies the algorithm’s implementation and reduces the computational resources required.
* The paper is well-structured, making the concepts accessible and the methodology easy to follow.

**Weaknesses:**

* The theoretical upper bound on the risk of the DPO is only weakly connected to the proposed SeRA. Additionally, the theoretical analysis does not account for the impact of multiple iterations on the algorithm's performance.

* The paper does not provide a thorough study into how the number of iterations affects SeRA's performance. While Figures 4 and 7 indicate that iter 3 consistently outperforms iter 2, it remains unclear whether further increasing the number of iterations would lead to additional performance improvements.

**Questions:**

* Is there an estimation of the uniform covering number for various hypothesis classes, such as linear functions and neural networks?

* Can the theoretical results presented in the paper be extended to apply to other SAA methods?

---

> ### Author Response · Authors · 2024-11-20
> **Response to Reviewer pPvp**
>
> We are very grateful for your constructive comments. We have expressed your concerns as we understood them and answered them as follows. If there is any concern that we may have misunderstood or if you have additional questions, we look forward to addressing them through further comments from you.
>
> **Q1. Takeaways from the theoretical perspective**
>
> > A1. The purpose of Remark 1 (and Theorem 1) is to provide theoretical intuition that selecting samples with a large IRM reduces over-optimization to spurious correlations by clearly distinguishing chosen and rejected responses. As noted in our manuscript, the analysis aims to show how large IRM values improve sample selection, rather than covering every aspect of the algorithm. To address the reviewer’s concerns, our response is divided into two parts:
>
> > 1. **Connection with SeRA**: Our theoretical result shows that selecting samples with a large IRM can lower the upper bound for misclassification of unseen samples, potentially leading to better generalization in distinguishing response pairs. This theoretical insight into the benefits of using IRM-based sample selection and preference data bootstrapping, manifests in the improved empirical performance of SeRA.
>
> > 2. **Consideration for multiple iterations**: While our theorem primarily addresses the theoretical advantage of IRM-based sample selection (bounded risk), it does not focus on iterative refinement. Using IRM over multiple iterations, as introduced in Eqn. (5), might introduce unexpected model bias and performance degradation (see Fig. 5), making theoretical guarantees difficult, which we empirically resolve by introducing an ensemble of IRMs across iterations (Eqn. (7)).
>
> **Q2. Experimental results for further iterations**
>
> > A2. To answer the question, we conducted additional experiments with further iterations, specifically from 2 to 5, evaluated on the UltraFeedback benchmark for both single and SFT performance. The experiments were performed on TinyLlama-1.1B with $k = 0.7N$ and $\tilde{k} = 0.3N$. The results are shown below:
>
> | Iteration    | Iter. 2 | Iter. 3 | Iter. 4 | Iter. 5 |
> |--------------|:-------:|:-------:|:-------:|:-------:|
> | Single       |   5.85  |   5.92  |   5.92  |   5.94  |
> | SFT (%)      |   70.2  |   73.1  |   74.5  |   74.8  |
>
> > The results suggest additional iterations may consistently improve performance (as performances for single grading are almost same across iterations 3 to 5), though the benefits diminish as iterations increase. We believe that future work could explore more scalable methods that continue to enhance performance as the number of iterations increases.
>
> **Q3. Discussion on uniform covering number for hypothesis classes**
>
> > A3. It is challenging to compute the exact values for uniform covering numbers directly. However, we can derive an upper bound under certain conditions [1, 2]. For example, let $H$ be the set of $L$-Lipschitz functions with respect to the infinity norm, mapping from $[0, 1]^d$ to $[0, 1]$. In this case, the (uniform) covering number, $N_\infty(\epsilon, H, N)$, will be a function of $(L/\epsilon)^d$ as defined in Definitions 1 and 2. For both linear functions and neural networks, we can define a Lipschitz constant using activation functions such as ReLU or GeLU, which ensures that this covering number is finite.
>
> > To further clarify this concept, consider that if two responses are easily distinguishable through sample selection, the complexity of the hypothesis class is reduced compared to the case without selection. **This is because the margin is larger, implying a smaller Lipschitz constant for the function class. Consequently, the (uniform) covering number is smaller with selection than without selection.**
>
> **Q4. Extension of theoretical results to other DAAs**
>
> > A4. Our theoretical results are specifically based on the fact that the implicit reward margin (IRM) can be directly represented in the logit values for probabilities in Bradley-Terry (BT) models within DPO. However, as noted in lines 348-350 in our manuscript, methods like IPO and SLiC-HF do not assume a preference model similar to BT, so the theoretical results in Remark 1 (or Theorem 1) cannot be directly extended to other DAAs.
> However, as demonstrated in Table 2 and Table 9, SeRA consistently performs well across IPO, SLiC-HF, and SimPO. Therefore, we believe it would be valuable future work to generalize our theoretical framework to apply to a broader range of DAA methods.
>
> [1] Asymptotic Statistics, Aad van der Vaart. Cambridge. 1998 \
> [2] Convergence of Stochastic Processes. David Pollard. Springer. 1984.

---

> ### Author Response · Authors · 2024-11-25
> **Gentle Reminder: Response Review for Reviewer pPvp**
>
> Dear Reviewer pPvp,
>
> In our response, we have addressed your questions by providing:
>
> - **Descriptions regarding the takeaways of Remark 1 (i.e., the connection with SeRA and considerations for multiple iterations)**,
> - **Additional experimental results for further training iterations**,
> - **An expanded description of the uniform covering number**, and
> - **An additional explanation regarding the potential extension of Remark 1**.
>
> We hope our response provides additional clarity regarding your questions. We would appreciate it if you could confirm that you have read our response and share any further questions you might have before the discussion period ends.
>
> Thank you for your time.
>
> Sincerely, \
> The Authors

---

> > ### Comment · Reviewer_pPvp · 2024-11-26
> >
> > Thank you for your response. I will keep my score.

---

> > > ### Author Response · Authors · 2024-11-28
> > > **Thank you for your response – Follow-Up on Concerns and Re-Evaluation for Reviewer pPvp**
> > >
> > > *Dear Reviewer pPvp,*
> > >
> > > Thank you for your response. We hope that our previous reply has adequately addressed all your concerns. If there are any additional questions or points you'd like us to clarify, please let us know before the discussion period ends.
> > >
> > > If all your concerns have been resolved, we would greatly appreciate it if you might consider re-evaluating your review.
> > >
> > > Sincerely, \
> > > The Authors

---

### Official Review · Reviewer_SFx5 · 2024-11-04

**Soundness:** 3
**Presentation:** 3
**Contribution:** 2
**Rating:** 6
**Confidence:** 5

**Summary:**

This paper studied the drawbacks of direct alignment algorithms that overfit the dataset and optimize the model to focus on the spurious correlations. To address these limitations, the authors proposed Self-Reviewing and Alignment which selects preference samples with high implicit reward margins.

**Strengths:**

1. The paper is well-written and easy to follow.
2. The experiments are extensive and well support the claims in the paper.

**Weaknesses:**

1. Since the DPO implicit reward can serve as a good approximation of the preference score, e.g., some of the top reward models in RewardBench [1] are from DPO implicit rewards, relabeling the preference data with DPO reparameterized reward is a natural way, and very much like the standard LLM-as-judge and RM-as-judge methods used to rank the preference pairs. In fact, similar approaches have been explored in [2,3].
2. The authors may argue that their method uses the implicit reward gap as a data selection metric, instead of only using the implicit reward for relabeling. However, given that implicit rewards are a form of reward, it might be the case that the data selection process mitigates the issues of DPO such as unlearning the high-quality chosen responses when the reward gap is small [2,4], which makes the performance improve. If this is true, then we may use any reward models to filter out similar-quality pairs, or use all preference data with the reward-aware algorithms proposed in [2,4].
3. The proposed method is computationally expensive since the implicit rewards need to be calculated for each data point, while the iterative DPO does not. Besides, given that the base models (e.g. the SFT models) outperform GPT 3.5 by a large margin and UltraFeedback uses GPT 3.5 for preference ranking, it might be the case that the reason that SeRA outperforms DPO is that the implicit reward is more accurate, even not removing any data. Therefore, I would like to see how iterative DPO performs when using DPO implicit reward to rank the pairs.
4. Based on the above comments, the evidence that SeRA is better than DPO since it addresses the spurious correlations issue may be weak.
5. It is also unclear what Theorem 1 indicates and how it relates to the filtering process of SeRA.

[1] Lambert et al., RewardBench: Evaluating Reward Models for Language Modeling.\
[2] Zhang et al., Reward-Augmented Data Enhances Direct Preference Alignment of LLMs.\
[3] Zhong et al., DPO Meets PPO: Reinforced Token Optimization for RLHF.\
[4] Nvidia, Nemotron-4 340B Technical Report.

**Questions:**

See weakness.

---

> ### Author Response · Authors · 2024-11-20
> **Response to Reviewer SFx5 (1/2)**
>
> We are very grateful for your constructive comments. We have expressed your concerns as we understood them and provided answers as follows. If there is any concern that we may have misunderstood or if you have additional questions, we would be happy to address them in further comments.
>
> **Q1. Difference from existing works**
>
> > A1. We would like to clarify that our work differs from existing approaches in three key ways:
>
> > 1. Our focus is on improving the efficiency of any direct alignment algorithms (DAAs) within a constrained cost budget, without relying on external reward models. We achieve this through sample selection and preference bootstrapping based on IRM of policy LLMs themselves rather than through simple relabeling. We believe that our goal significantly differs from the works mentioned by the reviewer, that either use external reward models [1] which require a lot re-labeling cost [2, 3] or provide token-wise reward to PPO training [4]. ($\color{magenta}{\text{Moreover, [1] was published on 10th October, 2024 on arxiv, which is after the paper submission deadline.}}$)
> > 2. We provide a rationale for the effectiveness of implicit rewards (IRM) through both empirical (Fig. 1 & Fig. 2) and theoretical (Remark 1) evidence, showing that samples with a large IRM can mitigate memorization of certain spurious correlations unrelated to the preference function such as longer response lengths.  **Moreover, our IRM-based sample selection has been shown to be more effective than other types of reward models, including GPT-4 or the log-probability of a reference model, as shown in Fig. 1.**
> > 3. Unlike previous approaches that focus solely on techniques like DPO or PPO, we propose a complementary approach that can be applied to any DAA and showcase its effectiveness against and alongside IPO, SLiC, and even SimPO which utilize different definitions of implicit reward terms.
>
> [1] Zhang et al., “Reward-Augmented Data Enhances Direct Preference Alignment of LLMs”. arXiv preprint. 2024 \
> [2] Rosset et al., “Direct Nash Optimization: Teaching Language Models to Self-Improve with General Preferences”. arXiv preprint. 2024 \
> [3] Guo et al., “Direct Language Model Alignment from Online AI Feedback”. arXiv preprint. 2024 \
> [4] Zhong et al., “DPO Meets PPO: Reinforced Token Optimization for RLHF.” ICML 2024 Workshop MFHAIA. 2024 \
>
> **Q2. Clarification for role of implicit reward margin**
> > A2. To address the reviewer's concerns, our response is two-fold:
>
> > 1. While our approach is in line with existing works mentioned by the reviewer, we have several important distinctions. First, we reiterate that we utilize the implicit reward margin both as a data selection metric for the offline dataset (Sec 3.2; lines 191-196) and for obtaining labeled pairs during data bootstrapping (Sec 3.3). Second, our reward computation considers an ensemble of models based on iterative improvement of the policy model. This approach is consistent with existing work, including those mentioned by the reviewers, as well as other selection baseline comparisons depicted in Fig. 1 and Fig. 2.
> > 2. As shown in Table 3 of our manuscript, we compared SeRA with a popular baseline using an external (Mistral-7B-based) reward model for TinyLlama-1.1B training (i.e., OAIF [3]). SeRA achieved 66.1% on HH-RLHF and 62.7% on TL;DR, outperforming OAIF, which scored 64.6% and 59.5%, respectively.
>
> > Overall, we believe our use of the implicit reward for both selection metrics and labeling is well-founded, supported by both theoretical intuition and empirical evidence, leading to strong performance in our experiments.
>
> **Q3. Addressing concerns regarding superiority of SeRA is from strong SFT models**
>
> > A3. To address the reviewer's concerns, our response is twofold:
>
> > 1. SeRA’s performance is not driven by strong base models. First, we want to clarify that $\color{magenta}{\text{both original and binarized versions of UltraFeedback labeling was done using GPT-4, not GPT-3.5 [5, 6].}}$ In contrast, our base models, as shown in the Alpaca Eval results in **Table 1** in the main manuscript for TinyLlama and Pythia SFT, are significantly weaker than GPT-4 and even GPT-3.5. Therefore, SeRA’s superiority cannot be attributed to the strength of its base models.
> > 2. We have already reported results using iterative DPO on UltraFeedback, which serves as a variant of SeRA without sample selection or bootstrapping (see Fig. 4, leftmost points of solid lines). For clarity, we summarize those results below:
>
> | WR (%)        | Iter. 2 | Iter. 3 |
> |:-------------:|:-------:|:-------:|
> | Iterative DPO |   59.4  |   64.7  |
> | SeRA          |   70.2  |   73.1  |
>
> > Based on these results, we believe that SeRA’s superiority does not stem from using stronger SFT models.
>
> [5] Cui et al., "UltraFeedback: Boosting Language Models with High-quality Feedback." ICML. 2024 \
> [6] HuggingFace., "https://huggingface.co/datasets/HuggingFaceH4/ultrafeedback_binarized". 2023

---

> ### Author Response · Authors · 2024-11-20
> **Response to Reviewer SFx5 (2/2)**
>
> **Q4. Highlight for effectiveness of IRM on spurious correlation**
>
> > A4. To further illustrate the relationship between IRM and spurious correlations, we evaluated the $R^2$ scores for GPT-4 scores vs. implicit rewards, as well as response length vs. implicit rewards, using varying selection ratios (0.6 to 1.0, in intervals of 0.1).
>
> | $R^2$ (Implicit Reward) |  0.6 |  0.7 |  0.8 |  0.9 |  1.0 |
> |-------------------------|:----:|:----:|:----:|:----:|:----:|
> | vs. GPT-4               | 0.07 | 0.08 | 0.08 | 0.08 | 0.08 |
> | vs. Length              | 0.05 | 0.08 | 0.10 | 0.11 | 0.13 |
>
> > From these results, we draw three conclusions:
>
> > Even with fewer training iterations at a selection ratio of 0.7, the $R^2$ score (vs. GPT-4) remains comparable to higher ratios (0.8 to 1.0).
> The $R^2$ score (vs. Length) consistently increases as the selection ratio and training iterations grow.
> For a selection ratio of 0.6, both $R^2$ scores are lower than for 0.7, indicating that the LLM has not yet adequately fit the dataset.
>
> > These findings support our claim that LLMs initially learn explicit features (i.e., true preferences) before gradually memorizing spurious correlations (e.g., response length). Our use of the IRM helps the model focus on true preferences while minimizing spurious correlations, highlighting IRM as a core component of SeRA in addressing this issue.
>
> **Q5. Clarification for Theorem 1**
>
> > A5. Theorem 1 provides mathematical support for the empirical phenomena in **Fig. 2**, motivating our use of IRM. It shows that selecting samples with a large IRM reduces over-optimization to spurious correlations by clearly distinguishing a chosen response from a rejected response. Precisely, Theorem 1 supports that training on clear preferences lowers the upper bound of the risk $f(x, y_w, y_l)$ compared to using datasets with ambiguous samples. This theoretical support for using IRM in sample selection for both off-policy and on-policy sampling manifests in the effectiveness of SeRA across DAAs, across datasets, etc.
>
> ## References
>
> [1] Zhang et al., “Reward-Augmented Data Enhances Direct Preference Alignment of LLMs”. arXiv preprint. 2024 \
> [2] Rosset et al., “Direct Nash Optimization: Teaching Language Models to Self-Improve with General Preferences”. arXiv preprint. 2024 \
> [3] Guo et al., “Direct Language Model Alignment from Online AI Feedback”. arXiv preprint. 2024 \
> [4] Zhong et al., “DPO Meets PPO: Reinforced Token Optimization for RLHF.” ICML 2024 Workshop MFHAIA. 2024 \
> [5] Cui et al., "UltraFeedback: Boosting Language Models with High-quality Feedback." ICML. 2024 \
> [6] HuggingFace., "https://huggingface.co/datasets/HuggingFaceH4/ultrafeedback_binarized". 2023

---

> ### Author Response · Authors · 2024-11-25
> **Gentle Reminder: Response Review for Reviewer SFx5**
>
> Dear Reviewer SFx5,
>
> In our response, we have addressed your questions by providing:
>
> - **A clear differentiation from existing works**,
> - **A clarification of the role of the IRM**,
> - **An additional explanation that strong SFT models are not the primary reason for SeRA's superiority**,
> - **A highlight of the effectiveness of the IRM in addressing spurious correlations**, and
> - **An additional clarification regarding Remark 1**.
>
> We hope our response provides additional clarity regarding your questions. We would appreciate it if you could confirm that you have read our response and share any further questions you might have before the discussion period ends.
>
> Thank you for your time.
>
> Sincerely, \
> The Authors

---

> ### Author Response · Authors · 2024-11-28
> **Kindly Following Up: Response Review for Reviewer SFx5**
>
> *Dear Reviewer SFx5,*
>
> We hope our response and revised manuscript provided additional information regarding your questions. We would appreciate it if you could acknowledge that you have read our response and share any further questions or concerns before the discussion period ends.
>
> If there are specific points you’d like us to clarify, please let us know, and we would be happy to elaborate to address your concerns.
>
> If there are no further questions, we hope we have satisfactorily addressed your concerns and would be delighted if you might consider raising in your evaluation.
>
> Sincerely, \
> The Authors

---

> ### Comment · Reviewer_SFx5 · 2024-12-03
>
> The authors' comments have addressed most of my concerns, and I have increased my score to 6. I hope the reviewers can incorporate the above discussions into the new iteration of the manuscript.

---

> > ### Author Response · Authors · 2024-12-03
> >
> > *Dear Reviewer SFx5,*
> >
> > Thank you for considering the reviewers' comments and adjusting the rating. We will review the updates to ensure the revisions have been fully addressed.
> >
> > Best, \
> > Authors

---

### Official Review · Reviewer_FEME · 2024-11-04

**Soundness:** 3
**Presentation:** 3
**Contribution:** 3
**Rating:** 6
**Confidence:** 3

**Summary:**

This work provide a method SeRA for Direct Alignment Algorithms (DAA). This method mainly uses the DAA policy as a reward model and provide implicit reward margins (IRM) for pairs. This IRM can be used for data selection and can also score the on-policy data to get more data. Many experiments are done to support the performance of this method.

**Strengths:**

1. This work is well-written and easy to read.

2. The proposed method is simple and don't need further human or AI annotation.

3. This method is suitable for different DAA methods, which makes it a quite general method.

4. The experimental results show the effectiveness.

**Weaknesses:**

1. As far as I can understand, I think this method mainly uses the DAA model as a reward model to generate on-policy signals. Therefore, I am quite curious about the comparison between SeRA and a direct DAA which uses on-policy preference data. For this direct DAA, the on-policy data might come from GPT-4 or a reward model trained from previous data.

2. I hope more intuition about that SeRA can mitigate over-optimization can be provided. Fig 2 shows that SeRA selected data have less dependency on response length but I am curious why this dependency can be removed.

3. I wonder if there are some other designs for r_t in Equ.(7), Sec. 3.4?

**Questions:**

See weakness part above.

---

> ### Author Response · Authors · 2024-11-20
> **Response to Reviewer FEME**
>
> We are very grateful for your constructive comments. We have rephrased your comments as we understand them and provided corresponding answers. Please let us know if there are any misunderstandings or if you have any additional questions.
>
> **Q1. Comparison performance between SeRA and on-policy training with external reward models**
>
> > A1. We previously reported a comparison between SeRA and OAIF [1], which uses on-policy training with external reward models (based on Mistral-7B), in Table 3. In these experiments, we evaluated TinyLlama-1.1B on the HH-RLHF [2] and TL;DR [3] datasets using a Mistral-7B-based reward model for on-policy training. As shown, SeRA achieved 66.1% on HH-RLHF and 62.7% on TL;DR, outperforming the on-policy counterpart which scored 64.6% and 59.5% respectively. These results demonstrate that SeRA can surpass on-policy training in this experimental setup.
>
> **Q2. Further explanation on how SeRA mitigates over-optimization**
>
> > A2. As noted in the main manuscript (lines 238-245), DNNs initially learn explicit features (e.g., preference) but later memorize spurious features (e.g., sequence length). In **Fig. 2**, while both the first (no selection) and fourth (IRM selection) columns show similar correlations between implicit rewards and true preferences (GPT-4 scores), the first column had a much higher $R^2$ correlation (0.13) compared to the fourth (0.08). This suggests that the larger sample size in the first column allows more memorization of spurious features due to extended training and encountering ambiguous preference samples. These findings align with existing findings that longer training increases correlation with sequence length [4]. To mitigate this, we focused on learning from more distinct pairs using larger IRM values to reduce the risk of learning spurious correlations from ambiguous pairs.
>
> > To further support our theoretical results, we evaluated the $R^2$ scores between GPT-4 scores vs. implicit rewards, as well as response length vs. implicit rewards, using varying selection ratios (0.6 to 1.0 in intervals of 0.1) to illustrate the relationship between IRM and spurious correlations.
>
> | $R^2$ (Implicit Reward) |  0.6 |  0.7 |  0.8 |  0.9 |  1.0 |
> |-------------------------|:----:|:----:|:----:|:----:|:----:|
> | vs. GPT-4               | 0.07 | 0.08 | 0.08 | 0.08 | 0.08 |
> | vs. Length              | 0.05 | 0.08 | 0.10 | 0.11 | 0.13 |
>
> > From these results, we draw three key conclusions:
>
> > 1. Even with fewer training iterations at a selection ratio of 0.7, the $R^2$ score (vs. GPT-4) remains comparable to higher ratios (0.8 to 1.0).
> > 2. The $R^2$ score (vs. Length) consistently increases as the selection ratio and training iterations grow.
> > 3. At a selection ratio of 0.6, both $R^2$ scores are lower than at 0.7, indicating that the LLM has not yet fully fit the dataset.
>
> > These findings support our claim that LLMs first learn explicit features (i.e., true preferences) before gradually memorizing spurious correlations (e.g., response length). Our use of IRM helps the model focus on true preferences while minimizing spurious correlations, supporting both mathematically and empirically, thereby justifying its use as a core component of our proposed method.
>
> **Q3. Any other potential design for $r_t$ in Eqn. (7)**
>
> > A3. An alternative for r_t ​ in Eqn. (7) is length-regularized (LR) reward shaping, which penalizes response length to reduce bias from implicit rewards [5]:
>
> $r_{\text{LR}}(\mathbf{x}, \mathbf{y}; \alpha) = \beta \log \frac{\pi_\theta(\mathbf{y} \mid \mathbf{x})}{\pi_{\text{ref}}(\mathbf{y} \mid \mathbf{x})} - \alpha |\mathbf{y}|$
>
> > In our comparison for the third iteration DPO with TinyLlama-1.1B, while LR reward achieved 79.4% and 67.6% for Vicuna and Evol-Instruct, respectively, our proposed method achieved 85.8% and 72.3%, demonstrating superior performance over the LR alternative.
>
> > We note that the distincts margins in our case can capture other spurious correlations beyond length where ambiguity over the preference can make reward models/humans consider other aspects (such as length, style of writing, etc.) to give preference annotations.
>
> [1] Guo et al., “Direct Language Model Alignment from Online AI Feedback.” arXiv preprint. 2024. \
> [2] Ganguli et al., “Red teaming language models to reduce harms: Methods, scaling behaviors, and lessons learned.” arXiv preprint. 2022. \
> [3] Stiennon et al., “Learning to summarize with human feedback.” NeurIPS. 2020 \
> [4] Rafailov et al., “Scaling laws for reward model overoptimization in direct alignment algorithms”. arXiv preprint. 2024 \
> [5] Chen et al., “Bootstrapping Language Models with DPO Implicit Rewards.” arXiv preprint. 2024

---

> ### Author Response · Authors · 2024-11-25
> **Gentle Reminder: Response Review for Reviewer FEME**
>
> Dear Reviewer FEME,
>
> In our response, we have addressed your question by providing:
>
> - **A comparison with on-policy training using an external reward model**
> - **An additional description of Remark 1**, and
> - **An additional comparison with alternatives for implicit reward design**.
>
> We hope our response provides additional clarity regarding your questions. We would appreciate it if you could confirm that you have read our response and share any further questions you might have before the discussion period ends.
>
> Thank you for your time.
>
> Sincerely, \
> The Authors

---

> ### Author Response · Authors · 2024-11-28
> **Kindly Following Up: Response Review for Reviewer FEME**
>
> *Dear Reviewer FEME,*
>
> We hope our response and revised manuscript provided additional information regarding your questions. We would appreciate it if you could acknowledge that you have read our response and share any further questions or concerns before the discussion period ends.
>
> If there are specific points you’d like us to clarify, please let us know, and we would be happy to elaborate to address your concerns.
>
> If there are no further questions, we hope we have satisfactorily addressed your concerns and would be delighted if you might consider reflecting this in your re-evaluation.
>
> Sincerely, \
> The Authors

---

> ### Author Response · Authors · 2024-12-03
> **A Gentle Final Reminder for Reviewer FEME**
>
> *Dear Reviewer FEME,*
>
> We sincerely hope our previous response has sufficiently addressed all your concerns. If you feel that your concerns have been resolved, we would be most grateful if you could kindly consider updating your evaluation accordingly.
>
> With best regards, \
> Authors

---

### Author Response · Authors · 2024-11-20
**General Response by Authors**

We sincerely thank all the reviewers (**@AqA6, @pPvp, @SFx5, @FEME**) for their careful review of our paper and for providing insightful and constructive feedback. We are pleased that the reviewers recognized the clear motivation behind our proposed method from both theoretical and empirical perspectives (**@AqA6, @pPvp**). We are glad to hear that (1) the reviewers found our paper well-written, well-structured, and easy to follow (**@pPvp, @SFx5, @FEME**) and (2) that the experimental results demonstrate significant improvements and effectively support the claims in our paper (**@AqA6, @pPvp, @SFx5, @FEME**). We have endeavored to address the concerns raised by the reviewers in our individual responses and have revised our manuscript accordingly based on their comments.

The following revisions have been made:

- Much clearer description of Remark 1 for @AqA6, @pPvp, @SFx5, @FEME (Section 3.2 & Appendix B)
- Additional literature review for @AqA6 (Appendix A)
- Clearer description of Algorithm 1 and benchmarking details for @AqA6 (Appendix C.1)
- Extension of Figure 2 across various selection ratios to further support our claims (Appendix D.4)
- Performance analysis for subcategories of the Evol-Instruct benchmark for @AqA6 (Appendix D.5)
- Discussion on the computational budget of SeRA for @SFx5 (Appendix D.6)
- Experimental results for additional iterations for @pPvp (Appendix D.7)

The revisions are marked in $\color{magenta}{\text{magenta}}$ in the updated paper. Additionally, we are happy to engage in further discussions during the rebuttal phase.

Thank you for your time and thoughtful feedback.

Sincerely,

The Authors

---

### Author Response · Authors · 2024-11-23
**Reminder: Response Review for All Reviewers**

Dear Reviewers,

Thank you for your thoughtful comments on our work. We have carefully addressed your initial feedback and provided corresponding responses, along with an updated manuscript.

We wanted to kindly follow up to see if there are any additional thoughts or clarifications you would like us to consider. Please let us know if there’s anything further we can do to clarify any points or address any concerns.

Thank you again for your time and efforts!

Sincerely,

The Authors

---

### Meta-Review · Area_Chair_mhKY · 2024-12-20

**Metareview:**

The paper proposed a framework that iterates between selecting samples based on the margin of the implicit reward model and training the model using the selected preference data. The paper claims that this approach addresses the over-optimization issue from offline DPO. The paper empirically demonstrates that their approach indeed outperforms baselines under multiple models and datasets.

**Additional Comments On Reviewer Discussion:**

The reviewers are in general positive about this paper. The concerns raised by the reviewers were around the clarification of the main theoretical result (i.e., remark 1) of the paper, comparison to online DPO based approaches that use external reward models, and related work on over-optimization in DPO. During the rebuttal, the authors successfully addressed these concerns which made corresponding reviewers raise their score and agree to accept.

---

### Decision · Program_Chairs · 2025-01-22

Accept (Poster)